# Lis1 regulates dynein by sterically blocking its mechanochemical cycle

**Katerina Toropova[1†], Sirui Zou[2†], Anthony J Roberts[2,3], William B Redwine[1,2], Brian S Goodman[2], Samara L Reck-Peterson[2]\*, Andres E Leschziner[1]\***

[1]Department of Molecular and Cellular Biology, Harvard University, Cambridge, United States; [2]Department of Cell Biology, Harvard Medical School, Boston, United States; [3]Astbury Centre for Structural Molecular Biology, University of Leeds, Leeds, United Kingdom

**Abstract** Regulation of cytoplasmic dynein's motor activity is essential for diverse eukaryotic functions, including cell division, intracellular transport, and brain development. The dynein regulator Lis1 is known to keep dynein bound to microtubules; however, how this is accomplished mechanistically remains unknown. We have used three-dimensional electron microscopy, single-molecule imaging, biochemistry, and in vivo assays to help establish this mechanism. The three-dimensional structure of the dynein–Lis1 complex shows that binding of Lis1 to dynein's AAA+ ring sterically prevents dynein's main mechanical element, the 'linker', from completing its normal conformational cycle. Single-molecule experiments show that eliminating this block by shortening the linker to a point where it can physically bypass Lis1 renders single dynein motors insensitive to regulation by Lis1. Our data reveal that Lis1 keeps dynein in a persistent microtubule-bound state by directly blocking the progression of its mechanochemical cycle.

**\*For correspondence:** reck-peterson@hms.harvard.edu (SLR); aleschziner@mcb.harvard.edu (AEL)

[†]These authors contributed equally to this work

**Competing interests:** The authors declare that no competing interests exist.

**Reviewing editor**: John Kuriyan, Howard Hughes Medical Institute, University of California, Berkeley, United States

## Introduction

Cytoplasmic dynein ('dynein' here), the largest and least understood of the cytoskeletal motors, uses the energy from ATP hydrolysis to move towards the minus ends of microtubules (*Vale, 2003*; *Carter, 2013*). As the major minus-end-directed motor in most eukaryotic cells, dynein's many roles include transporting a range of macromolecular cargo (*Blocker et al., 1997*; *Jordens et al., 2001*; *Kural et al., 2005*; *Pilling et al., 2006*; *Driskell et al., 2007*), constructing and positioning the mitotic spindle (*Heald et al., 1996*; *Merdes et al., 1996*; *Kiyomitsu and Cheeseman, 2013*), and polarizing and anchoring mRNAs during development (*Wilkie and Davis, 2001*). To perform its diverse biological functions, dynein partners with a range of regulatory co-factors, an important subset of which can alter dynein motility directly. Despite progress in understanding the architecture and mechanism of dynein's large motor domain, how this structure is acted upon by regulatory factors is not yet known.

Dynein is a homodimer of force generating units (~500 kDa each) (*Figure 1A,B*). The N-terminal region of each monomer forms the 'tail' domain, which mediates dimerization and cargo attachment via adaptor proteins. Removal of the tail yields the 'motor', the minimal portion of dynein that can exert force. At the core of the motor are six AAA+ modules (AAA1–6) that fold into a ring. AAA1 is the main site of ATP hydrolysis for motility but AAA2, 3, and 4 can also bind ATP, and AAA3 and 4 can hydrolyze it (*Gibbons et al., 1991*; *Kon et al., 2004, 2012*; *Cho et al., 2008*; *Schmidt et al., 2012*). AAA5 and AAA6 have lost the ability to bind nucleotide (*Kon et al., 2012*; *Schmidt et al., 2012*). Two appendages to the ring are essential for dynein function; the 'stalk', an intramolecular anti-parallel coiled-coil at the end of which lies the microtubule-binding domain (*Gee et al., 1997*; *Carter et al., 2008*) and the 'linker', which is dynein's key mechanical element and is an elongated structure N-terminal to AAA1. The linker spans the ring and moves in a nucleotide dependent manner that is

**eLife digest** Cells use motor proteins to move 'cargo' from one location to another inside the cell. This cargo can range in size from a single macromolecule to something as large as the nucleus of the cell. A motor protein called dynein is the largest and least understood of the motor proteins found in cells.

Dynein molecules work in pairs to take 'steps' along tracks called microtubules. Dynein contains two domains: a motor domain, which is responsible for generating movement, and a 'tail' domain to which the cargo is attached. The motor domain is composed of a ring-like shape and two appendages—the stalk and the linker. The linker undergoes large-scale movements relative to the ring that transmits force to the tail domain.

Dynein also interacts with various accessory proteins to do its job inside the cell. One of these is a protein called Lis1 that is found across a wide range of species from yeast to humans. Defects in the gene for Lis1 result in brain developmental disorders in humans. However, it is not clear how the Lis1 protein influences the activity of dynein.

Now Toropova, Zou et al. have visualized the structure of dynein bound to Lis1 and compared it with the structure of dynein on its own in order to work out if dynein changes its shape as a result of binding to Lis1. These experiments show that when Lis1 binds to dynein, it physically blocks the linker, preventing it from making contacts with the ring-like shape that are important for the normal function of the motor.

To test the idea that this physical block is responsible for dynein molecules spending a relatively long time attached to their microtubules, Toropova, Zou et al. shortened the linker to a point where the Lis1 protein could no longer block it: this resulted in a dynein motor that was no longer sensitive to Lis1. A challenge for the future is to understand, at a molecular level, how the Lis1-mediated slowing down of dynein affects the multiple functions the motor carries out in a cell.

thought to transmit force to dynein's cargo (*Burgess et al., 2003*; *Kon et al., 2005*; *Shima et al., 2006*; *Roberts et al., 2009, 2012*). In order for dynein to move along microtubules, ATP binding/hydrolysis at AAA1 must be coupled with linker motion and microtubule binding and release at the tip of the stalk, located 250 Å away (*Gibbons et al., 2005*; *Imamula et al., 2007*; *Kon et al., 2009*; *Redwine et al., 2012*).

Across a wide range of species, dynein interacts with a conserved regulator called Lis1 (also known as Pac1 in *Saccharomyces cerevisiae*) that is necessary for many dynein driven processes. Mutations in the Lis1 gene cause the neurodevelopmental disorder lissencephaly (*Reiner et al., 1993*). Lis1 is the only dynein regulator known to interact directly with its motor domain (*McKenney et al., 2010*; *Huang et al., 2012*). Like dynein, Lis1 acts as a dimer, with each monomer comprising an N-terminal dimerization domain (LisH) followed by a coiled-coil, a flexible loop and a C-terminal β-propeller domain of 7 WD motifs (*Kim et al., 2004*; *Tarricone et al., 2004*) (*Figure 1A,B*). We previously showed that the propeller domain alone can regulate dynein in vitro and used negative stain electron microscopy (EM) and two-dimensional (2D) image processing to show that Lis1 binds to dynein's motor domain near AAA3/4 (*Huang et al., 2012*). We and others have shown that Lis1 induces a slow-moving microtubule-attached state in dynein (*Yamada et al., 2008*; *McKenney et al., 2010*; *Torisawa et al., 2011*; *Huang et al., 2012*). Interestingly, Lis1 can accomplish this without substantially affecting dynein's overall ATP hydrolysis rate (*Yamada et al., 2008*; *McKenney et al., 2010*; *Huang et al., 2012*). This led us to propose that Lis1 acts as a 'clutch', uncoupling dynein's engine (AAA+ ring) from its track-binding region.

Our previous 2D EM data, which indicated that Lis1 binds in the vicinity of AAA3/4, raised at least two possibilities for how Lis1 can affect dynein's mechanochemistry. On the one hand, Lis1 may regulate dynein allosterically, influencing the structure or motions of AAA3 or AAA4 in the ring, and thus preventing the propagation of a signal for microtubule detachment to the stalk. Alternatively, because the linker domain lies close to AAA4 in certain nucleotide-states, Lis1 may regulate dynein sterically, affecting the linker's movement directly, and thus dynein's mechanochemistry. Establishing Lis1's mode of regulation is not possible without three-dimensional (3D) data, as it is not known if Lis1 binds on the same face of the AAA+ ring as the linker, as would be required for direct Lis1–linker interactions

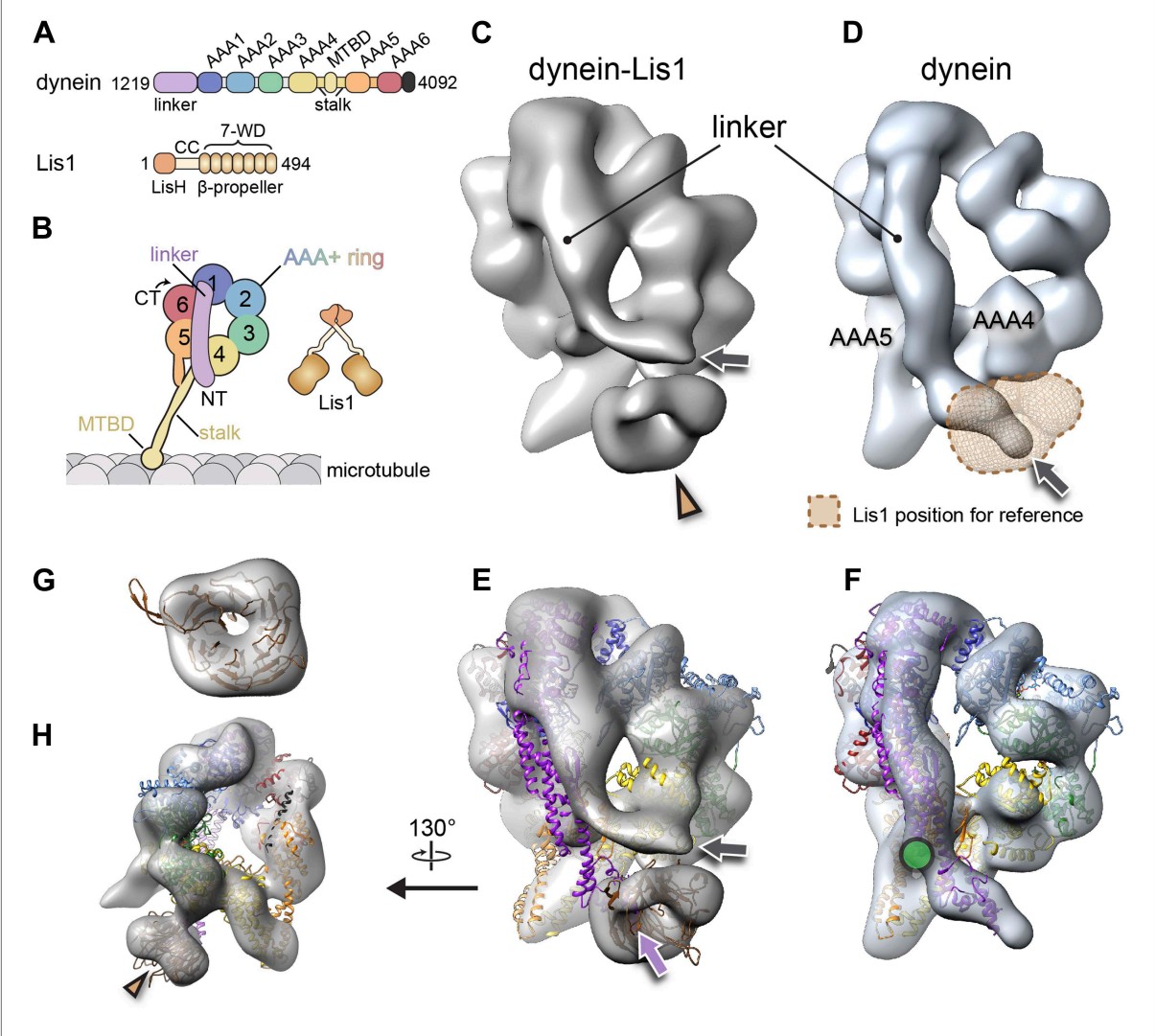

**Figure 1**. The binding of Lis1 to dynein changes the position of dynein's linker domain. (**A**) Domain organization of dynein and Lis1 constructs used in this study. Dynein's AAA+ domains are labeled AAA1–6. MTBD: microtubule binding domain; CC: coiled coil; LisH: Lis-homology (dimerization) motif. (**B**) Schematic representation of dynein and Lis1, color-coded as in (**A**) and throughout the paper. NT: N terminus; CT: C terminus. (**C**) Cryo-NS EM reconstruction of the dynein motor domain in complex with Lis1 and (**D**) of the motor domain alone. AAA4 and AAA5 are labeled. A density present only in the dynein–Lis1 map is highlighted in (**C**) (brown arrowhead). The linker occupies different positions in the two maps (compare labeled densities and gray arrows), and its position in the dynein alone map is sterically incompatible with Lis1, as indicated by a semi-transparent Lis1 density. (**E**) Structural model of dynein's motor domain docked into the EM maps of dynein–Lis1 and (**F**) dynein alone. The model was built from crystal structures of the *S. cerevisiae* dynein ring (PDB ID: 4AKG [***Schmidt et al., 2012***]) and *D. discoideum* linker aligned to the yeast linker position (PDB ID: 3VKG [***Kon et al., 2012***]), the *D. discoideum* linker being closer in length to that in our EM construct. In (**E**), a homology model of the *S. cerevisiae* Lis1 β-propeller (brown) has been docked into the new density highlighted in (**C**). The linker domain in the EM map (gray arrow) is shifted away from its position in the crystal structure (purple arrow), which protrudes from the EM density and clashes with the Lis1 density. In contrast, the linker is within the EM density in the dynein alone map (**F**). Green circle: location of known interactions between the linker and AAA5 module in dynein (***Schmidt et al., 2012***). (**G**) Close-up view of the Lis1 density with homology model docked in, viewed along the axis indicated by the arrowhead in (**C**). (**H**) A rotated, smaller view of (**E**), showing the interface between Lis1 (brown arrowhead) and dynein.

The following figure supplements are available for figure 1:

**Figure supplement 1**. Three-dimensional (3D) classification and refinement of the dynein and dynein–Lis1 reconstructions.

**Figure supplement 2**. The linker's displaced position in the presence of Lis1 does not appear to involve a specific interaction with AAA4.

to occur. Moreover, it is not clear to what extent the Lis1 binding site encompasses AAA3, AAA4, or both of these modules. Thus, 3D structural information is critical to understanding the mechanistic basis of Lis1's regulation of dynein.

We set out to establish how Lis1 induces a persistent microtubule-bound state in dynein. We obtained the 3D structure of *S. cerevisiae* dynein bound to Lis1 to determine which elements of the motor Lis1 directly affects. Our structure revealed that Lis1 sterically prevents the linker from reaching its normal post-powerstroke locations on the ATP hydrolyzing ring that are involved in its conformational cycle. Structure-based mutagenesis also allowed us to identify residues in Lis1 responsible for binding to dynein. Single molecule analysis of a dynein motor with a shortened linker that can physically bypass Lis1 indicated that removing the steric block renders dynein insensitive to Lis1. Our combined data show that Lis1 directly blocks dynein's mechanochemical cycle, inducing a persistent microtubule-bound state, by acting on its linker domain.

## Results

### Structure of the dynein–Lis1 complex

In order to visualize the spatial relationship between Lis1 and dynein's multiple structural elements and to better understand the mechanism by which Lis1 regulates dynein, we used cryo-negative stain (cryo-NS) EM and single-particle image processing to obtain the 3D structure of the dynein–Lis1 complex (*Figure 1C*). Cryo-NS combines the structural preservation of vitrification with the high contrast provided by the negative stain (*De Carlo and Stark, 2010*). We found this increased contrast to be instrumental to our ability to computationally sort the different conformations that co-existed in most of our samples. We also determined a 3D map of dynein alone, as a reference, to establish whether Lis1 alters dynein's structure (*Figure 1D*). We used a well-characterized monomeric dynein construct (*Reck-Peterson et al., 2006*) but chose to use dimeric rather than monomeric Lis1 for our reconstructions. We previously showed that while a Lis1 monomer is sufficient to slow down dynein, a much higher concentration of it is required (*Huang et al., 2012*), presumably due to the high local concentration of the β-propeller in the context of a Lis1 dimer. Using a Lis1 dimer at much lower concentrations allowed us to minimize the background in our images.

Both dynein and Lis1 were expressed from *S. cerevisiae* at their genomic loci (*Table 1*). We imaged dynein–Lis1 and dynein alone in the absence of nucleotide and obtained structures at resolutions of 21 Å for the complex (*Figure 1C,E*, *Figure 1—figure supplement 1C* and *Video 1*) and of 15 Å for dynein alone (*Figure 1D,F*, *Figure 1—figure supplement 1C* and *Video 2*). The dynein alone map accommodates the crystal structures of the dynein motor domain well (*Kon et al., 2012*; *Schmidt et al., 2012*) (*Figure 1F*), with a Fourier Shell Correlation between the EM map and yeast motor domain structure (*Schmidt et al., 2012*) of 0.143 at a resolution of 18.8 Å.

The dynein–Lis1 map shows two major differences relative to the dynein alone reconstruction. First, a prominent donut-shaped density is resolved in contact with the dynein ring, adjacent to the stalk (*Figure 1C*, brown arrowhead). This extra density matches the dimensions of a β-propeller, including the hole at its center (*Figure 1E,G,H*). We thus conclude that the density corresponds to Lis1. The second, and striking difference between the two maps is in dynein itself: Lis1 binds on the same face of the ring where dynein's linker domain is located and the linker is displaced by ~44 Å in the dynein–Lis1 map relative to the dynein alone reconstruction (*Figure 1C–F*).

### One dynein ring binds one Lis1 β-propeller

Our previous 2D image analysis of the dynein–Lis1 complex did not allow us to determine whether their interaction involved one or both of Lis1's β-propellers or whether Lis1's N-terminal LisH dimerization domain was part of the interaction as well. The extra density in the dynein–Lis1 3D map fits well a single homology model of the *S. cerevisiae* Lis1 β-propeller built from the crystal structure of the mouse protein (*Tarricone et al., 2004*) (*Figure 1E,G,H*). Because our map resolved the hole at the center of the propeller, the homology model could be unambiguously docked within the density in terms of its translation (*Figure 1G*).

In Lis1, the β-propeller is connected to the N-terminal LisH dimerization domain by a loop, predicted to be flexible, and a coiled coil (*Kim et al., 2004*; *Tarricone et al., 2004*). Consequently, the rest of Lis1 would be expected to adopt a wide range of positions relative to the dynein-bound propeller domain. In agreement with this, we did not resolve density beyond that of the single β-propeller

**Table 1.** Yeast strains

| Strain | Genotype | Figure(s) / Reference |
|---|---|---|
| RPY753 | MATa, *his3-11,15, ura3-1, leu2-3,112, ade2-1, trp1-1, pep4Ä::HIS5, prb1Ä, P$_{GAL1}$-ZZ-Tev-GFP-3xHA-GST-DYN1$_{331kDa}$-gs-DHA, pac1Ä::URA3, ndl1Ä::cgLEU2* | *Figure 2, Figure 2—figure supplement 1,2, Figure 5—figure supplement 1* |
| | | *Huang et al., 2012* |
| RPY816 | MATa, *his3-11,15, ura3-1, leu2-3,112, ade2-1, trp1-1, pep4Ä::HIS5, prb1Ä, P$_{GAL1}$-ZZ-Tev-PAC1, dyn1Ä::cgLEU2, ndl1Ä::Hygro$^R$* | *Figures 1–5, Figure 2—figure supplement 1,2, Figure 1—figure supplement 1, Figure 4—figure supplement 1, Figure 5—figure supplement 1* |
| | | Julie Huang, Harvard Medical School |
| RPY842 | MATa, *his3-11,15, ura3-1, leu2-3,112, ade2-1, trp1-1, pep4Ä::HIS5, prb1Ä, P$_{GAL1}$-ZZ-Tev-PAC1-g-1xFLAG-ga-SNAP-Kan$^R$, dyn1Ä::cgLEU2, ndl1Ä::Hygro$^R$* | *Figures 3,5, Figure 3—figure supplement 1, Figure 5—figure supplement 1* |
| | | *Huang et al., 2012* |
| RPY844 | MATa, *his3-11,15, ura3-1, leu2-3,112, ade2-1, trp1-1, pep4Ä::HIS5, prb1Ä, PAC11-13xMYC-TRP1, P$_{GAL1}$-ZZ-Tev-GFP-3xHA-DYN1$_{331kDa}$, pac1Ä::Hygro$^R$* | *Figures 1,4, Figure 1—figure supplement 1, Figure 3—figure supplement 1* |
| | | *Huang et al., 2012* |
| RPY1198 | MATa, *his3-11,15, ura3-1, leu2-3,112, ade2-1, trp1-1, pep4Ä::HIS5, prb1Ä, PAC11-13xMYC-TRP1, P$_{GAL1}$-ZZ-Tev-GFP-3xHA-DYN1$_{331kDa}$-gs-DHA-Kan$^R$, pac1Ä::Hygro$^R$* | *Figure 5, Figure 5—figure supplement 1* |
| | | *Huang et al., 2012* |
| RPY1245 | MATa, *ura3-52, lys2-801, leu2-Ä1, his3-Ä200, trp1-Ä63, SPC110-GFP::TRP1, HXT1-tdTomato::HIS3* | *Figure 2* |
| | | Jeff Moore, University of Colorado |
| RPY1248 | MATa, *ura3-52, lys2-801, leu2-Ä1, his3-Ä200, trp1-Ä63, SPC110-GFP::TRP1, HXT1-tdTomato::HIS3, dyn1Ä::URA3* | *Figure 2* |
| | | This work |
| RPY1302 | MATa, *his3-11,15, ura3-1, leu2-3,112, ade2-1, trp1-1, pep4Ä::HIS5, prb1Ä, PAC11-13xMYC-TRP1, P$_{GAL1}$-ZZ-Tev-DYN1$_{331kDa}$, pac1Ä::Hygro$^R$* | *Figures 1,3* |
| | | This work |
| RPY1400 | MATa, *his3-11,15, ura3-1, leu2-3,112, ade2-1, trp1-1, pep4Ä::HIS5, prb1Ä, PAC11-13xMYC-TRP1, P$_{GAL1}$-ZZ-Tev-GFP-3xHA-DYN1$_{331kDa}$-L2441ybbR, pac1Ä::Hygro$^R$* | *Figure 3, Figure 3—figure supplement 1* |
| | | This work |
| RPY1422 | MATa, *his3-11,15, ura3-52, leu2-3,112, ade2-1, trp1-1, pep4Ä::HIS5, prb1Ä, P$_{GAL1}$-ZZ-Tev-GFP-3xHA-DYN1$_{314kDa}$-gs-DHA, pac1Ä::Hygro$^R$* | *Figures 4,5, Figure 4—figure supplement 1, Figure 5—figure supplement 1* |
| | | This work |
| RPY1436 | MATa, *his3-11,15, ura3-52, leu2-3,112, ade2-1, trp1-1, pep4Ä::HIS5, prb1Ä, PAC11-13xMYC-TRP1, P$_{GAL1}$-ZZ-Tev- DYN1$_{314kDa}$, pac1Ä::Hygro$^R$* | *Figure 5* |
| | | This work |
| RPY1439 | MATa, *his3-11,15, ura3-1, leu2-3,112, ade2-1, trp1-1, pep4Ä::HIS5, prb1Ä, P$_{GAL1}$-ZZ-Tev-GFP-3xHA-GST-DYN1$_{314 kDa}$-gs-DHA-Kan$^R$, pac1Ä:URA3, ndl1Ä::cgLEU2* | *Figure 5—figure supplement 1* |
| | | This work |
| RPY1509 | MATa, *his3-11,15, ura3-1, leu2-3,112, ade2-1, trp1-1, pep4Ä::HIS5, prb1Ä, PAC11-13xMYC-TRP1, P$_{GAL1}$-ZZ-Tev-DYN1$_{331kDa}$-gs-DHA-Kan$^R$, pac1Ä::Hygro$^R$* | *Figure 5—figure supplement 1* |
| | | This work |
| RPY1510 | MATa, *his3-11,15, ura3-1, leu2-3,112, ade2-1, trp1-1, pep4Ä::HIS5, prb1Ä, PAC11-13xMYC-TRP1, P$_{GAL1}$-ZZ-Tev-DYN1$_{314kDa}$-gs-DHA-Kan$^R$, pac1Ä::Hygro$^R$* | *Figure 5—figure supplement 1* |
| | | This work |
| RPY1523 | MATa, *ura3-52, lys2-801, leu2-Ä1, his3-Ä200, trp1-Ä3, SPC110-GFP::TRP1, HXT1-tdTomato::HIS3, pac1Ä::URA3* | *Figure 2* |
| | | This work |
| RPY1524 | MATa, *ura3-52, lys2-801, leu2-Ä1, his3-Ä200, trp1-Ä63, SPC110-GFP::TRP1, HXT1-tdTomato::HIS3, PAC1$^{R378A}$* | *Figure 2* |
| | | This work |

*Table 1. Continued on next page*

*Table 1. Continued*

| Strain | Genotype | Figure(s)<br>Reference |
|---|---|---|
| RPY1525 | MATa, *ura3-52, lys2-801, leu2-Ä1, his3-Ä200, trp1-Ä63, SPC110-GFP::TRP1, HXT1-tdTomato::HIS3, PAC1$^{R275Ä,R301A,R378A,W419A,K437A}$* | *Figure 2*<br>This work |
| RPY1543 | MATa, *his3-11,15, ura3-1, leu2-3,112, ade2-1, trp1-1, pep4Ä::HIS5, prb1Ä, P$_{GAL1}$-ZZ-Tev-PAC1$^{R275Ä}$, dyn1Ä::cgLEU2, ndl1Ä::Hygro$^R$* | *Figure 2—figure supplement 1*<br>This work |
| RPY1544 | MATa, *his3-11,15, ura3-1, leu2-3,112, ade2-1, trp1-1, pep4Ä::HIS5, prb1Ä, P$_{GAL1}$-ZZ-Tev-PAC1$^{R378Ä}$, dyn1Ä::cgLEU2, ndl1Ä::Hygro$^R$* | *Figure 2, Figure 2—figure supplement 1,2*<br>This work |
| RPY1545 | MATa, *his3-11,15, ura3-1, leu2-3,112, ade2-1, trp1-1, pep4Ä::HIS5, prb1Ä, P$_{GAL1}$-ZZ-Tev-PAC1$^{W419A}$, dyn1Ä::cgLEU2, ndl1Ä::Hygro$^R$* | *Figure 2—figure supplement 1*<br>This work |
| RPY1546 | MATa, *his3-11,15, ura3-1, leu2-3,112, ade2-1, trp1-1, pep4Ä::HIS5, prb1Ä, P$_{GAL1}$-ZZ-Tev-PAC1$^{K437A}$, dyn1Ä::cgLEU2, ndl1Ä::Hygro$^R$* | *Figure 2—figure supplement 1*<br>This work |
| RPY1547 | MATa, *his3-11,15, ura3-1, leu2-3,112, ade2-1, trp1-1, pep4Ä::HIS5, prb1Ä, P$_{GAL1}$-ZZ-Tev-PAC1$^{R275A,R301A,R378A,W419A,K437A}$, dyn1Ä::cgLEU2, ndl1Ä::Hygro$^R$* | *Figure 2, Figure 2—figure supplement 1,2*<br>This work |
| RPY1548 | MATa, *his3-11,15, ura3-1, leu2-3,112, ade2-1, trp1-1, pep4Ä::HIS5, prb1Ä, P$_{GAL1}$-ZZ-Tev-PAC1$^{R301A}$, dyn1Ä::cgLEU2, ndl1Ä::Hygro$^R$* | *Figure 2—figure supplement 1*<br>This work |
| RPY1553 | MATa, *his3-11,15, ura3-1, leu2-3,112, ade2-1, trp1-1, pep4Ä::HIS5, prb1Ä, PAC11-13xMYC-TRP1, P$_{GAL1}$-ZZ-Tev-GFP-3xHA-DYN1$_{331kDa}$$^{E1849Q}$, pac1Ä::Hygro$^R$* | *Figure 4, Figure 4—figure supplement 1*<br>This work |
| RPY1554 | MATa, *his3-11,15, ura3-1, leu2-3,112, ade2-1, trp1-1, pep4Ä::HIS5, prb1Ä, PAC11-13xMYC-TRP1, P$_{GAL1}$-ZZ-Tev-GFP-3xHA-DYN1$_{331kDa}$$^{E2819Q}$, pac1Ä::Hygro$^R$* | *Figure 4, Figure 4—figure supplement 1*<br>This work |
| RPY1555 | MATa, *his3-11,15, ura3-52, leu2-3,112, ade2-1, trp1-1, pep4Ä::HIS5, prb1Ä, P$_{GAL1}$-ZZ-Tev-GFP-3xHA-DYN1$_{314kDa}$$^{K3438E,R3445E,F3446D}$-gs-DHA, pac1Ä::Hygro$^R$* | *Figure 4—figure supplement 1,*<br>*Figure 5—figure supplement 1*<br>This work |
| RPY1557 | MATa, *his3-11,15, ura3-1, leu2-3,112, ade2-1, trp1-1, pep4Ä::HIS5, prb1Ä, PAC11-13xMYC-TRP1, P$_{GAL1}$-ZZ-Tev-GFP-3xHA-DYN1$_{331kDa}$$^{K3438E,R3445E,F3446D}$_gs-DHA-Kan$^R$, pac1Ä::Hygro$^R$* | *Figure 4, Figure 4—figure supplement 1*<br>This work |
| RPY1623 | MATa, *his3-11,15, ura3-1, leu2-3,112, ade2-1, trp1-1, pep4Ä::HIS5, prb1Ä, P$_{GAL1}$-ZZ-Tev-GFP-3xHA-GST- DYN1$_{331kDa}$$^{R2857A,N2858A,K2859A,R2861A,S2862A}$_gs-DHA, pac1Ä::URA3, ndl1Ä::cgLEU2* | *Figure 1—figure supplement 2*<br>This work |

*DYN1, PAC11, PAC1, and NDL1* encode the dynein heavy chain, dynein intermediate chain, Lis1 and Nudel orthologs, respectively. *DHA, SNAP,* and *ybbR* refer to the HaloTag (Promega), SNAP-tag (NEB), and ybbR tag (**Yin et al., 2005**), respectively. *TEV* indicates a Tev protease cleavage site. *P$_{GAL1}$* denotes the galactose promoter, which was used for inducing strong expression of Lis1 and dynein motor domain constructs. Genes encoding proteases Pep4 and Prb1 were deleted as noted. Amino acid spacers are indicated by *g* (glycine), *ga* (glycine-alanine), and *gs* (glycine-serine).

in our dynein–Lis map. 2D image analysis of dynein–Lis1 complexes with either monomeric or dimeric Lis1 showed the same density and location for Lis1 (*Figure 1—figure supplement 1D,E*), further supporting a stoichiometry of one Lis1 propeller to one dynein motor domain.

## Dissection of the dynein–Lis1 interface

Our structure of the dynein–Lis1 complex shows that the Lis1 β-propeller contacts dynein primarily at a surface-exposed helix at the junction of AAA3 and AAA4 (*Figure 2A* and *Video 3*), explaining why

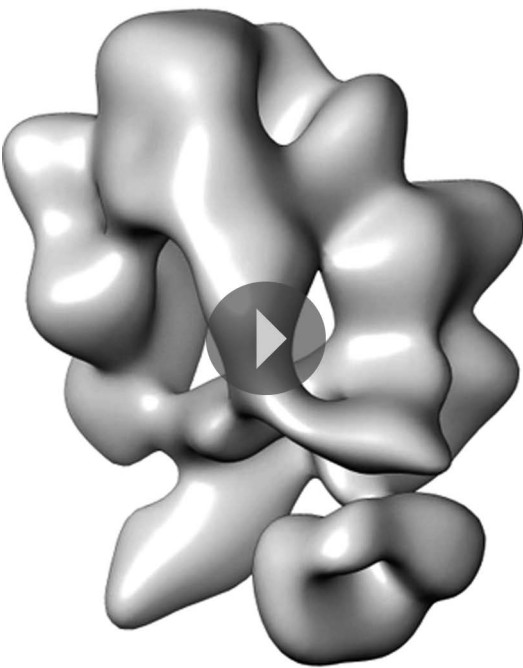

**Video 1**. The three-dimensional structure of dynein–Lis1. The movie shows the 3D reconstruction of dynein in complex with Lis1 with 360° rotation about the Y-axis. After this rotation, the EM density is made transparent to display the docked dynein crystal structure model and Lis1 homology model and is again rotated by 360° about the Y-axis.

mutagenesis of four conserved, charged residues (KDEE) on this helix virtually abolished Lis1 binding and dynein regulation (*Huang et al., 2012*). Since the resolution of the reconstruction does not allow us to determine unambiguously the rotational orientation of the Lis1 homology model within the corresponding density, we used mutagenesis to probe the dynein–Lis1 interface and further constrain our model of the complex.

Within Lis1, sequence conservation is much greater on one face of the β-propeller ('top') compared to the other (*Figure 2—figure supplement 1A*), suggesting that this top face may interact with dynein. Consistent with this idea, the docked β-propeller showed a better qualitative fit and a slightly higher cross-correlation coefficient with our density map when the top face is placed at the dynein interface (*Figure 2—figure supplement 1B*). To test this docking orientation, we mutated highly conserved residues on the top face of the propeller (*Figure 2B*). Our previous finding that the KDEE residues in dynein are critical for Lis1 binding (*Huang et al., 2012*) suggested that interactions between Lis1 and dynein have an electrostatic component. We therefore targeted four positively charged residues on the top propeller face, as well as a surface tryptophan, all of which are conserved (*Figure 2B*). We mutated these residues to alanine, both singly and in combination.

We first used size-exclusion chromatography to test the ability of the Lis1 mutants to interact with a functional dimerized dynein construct (GST-dynein$_{331kDa}$) (*Reck-Peterson et al., 2006*). When all five residues are mutated to alanine (Lis1[5A]), no binding could be detected by size-exclusion chromatography (*Figure 2C* and *Table 2*). We also did not detect an interaction with two of the single point mutants, Lis1[R378A] and Lis1[W419A] (*Figure 2C* and *Figure 2—figure supplement 1D*). The remaining single point mutants showed decreased but detectable binding to dynein (*Table 2* and *Figure 2—figure supplement 1D*). Thus, we conclude that highly conserved amino acids on the top face of Lis1's β-propeller are critical for dynein binding, in support of our structural model for the dynein–Lis1 complex (*Figure 2A*).

We next examined if the binding-deficient Lis1 mutants Lis1[5A] and Lis1[R378A] were able to regulate dynein in vitro. Wild-type Lis1 decreases dynein velocity in vitro in a concentration-dependent manner (*Huang et al., 2012*). These assays, where the motion of single, fluorescently labeled dynein molecules along microtubules is monitored over time, are more sensitive than size-exclusion chromatography for detecting dynein–Lis1 interactions. Therefore, we expected that some of the Lis1 mutants that did not co-migrate with dynein might still exhibit weak but measurable regulation of the motor. The Lis1[5A] mutant showed no reduction in dynein velocity, consistent with an inability to bind dynein (*Figure 2D* and *Figure 2—figure supplement 2*). The Lis1[R378A] mutant, on the other hand, showed a slight reduction in dynein velocity compared to dynein alone, suggesting that its binding to the motor is compromised (*Figure 2D* and *Figure 2—figure supplement 2*). Thus, the effect of the Lis1 mutations on dynein binding correlates with the ability of the Lis1 mutants to regulate dynein at the single-molecule level.

Lastly, we tested our model for the dynein–Lis1 complex by measuring the effect of disrupting the dynein–Lis1 interface in vivo. In yeast, spindle pole bodies (SPB) span the nuclear envelope and coordinate microtubule minus ends that emanate from its nuclear and cytoplasmic faces (*Jaspersen and Winey, 2004*). Lis1 assists in concentrating dynein at the plus ends of cytoplasmic microtubules, from where dynein is offloaded to the cell cortex (*Lee et al., 2003*; *Sheeman et al., 2003*; *Roberts et al., 2014*). Cortically anchored dynein exerts a pulling force that results in displacements of the entire

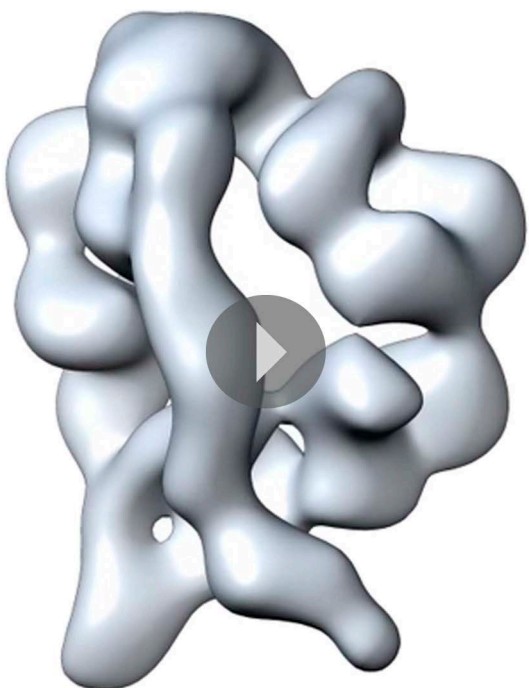

**Video 2**. The three-dimensional structure of dynein. The movie shows the 3D reconstruction of dynein alone with 360° rotation about the Y-axis. After this rotation, the EM density is made transparent to display the docked dynein crystal structure model and is again rotated by 360° about the Y-axis.

mitotic spindle (*Moore et al., 2009*), giving rise to a brief series of oscillations across the bud neck. Deletion of dynein eliminates these oscillations (*Yeh et al., 1995*). We quantified the effect of the Lis1[5A] and Lis1[R378A] mutants on spindle movement in cells treated with hydroxyurea, which prolongs the period of oscillations, by tracking fluorescently labeled SPBs over the course of 20 min. We found that disruption of the dynein–Lis1 interface resulted in a decrease in the number of bud neck crossings to a level similar to that caused by the deletion of Lis1 (*Figure 2E*). These results indicate that the dynein–Lis1 interface identified in our structural model is crucial for dynein's biological function.

## Lis1 sterically blocks the position adopted by the linker under ADP and no nucleotide conditions but does not prevent it from reaching the pre-powerstroke position at AAA2

The end of the linker domain is displaced ~44 Å in the dynein–Lis1 structure relative to the dynein alone map, mainly along the plane of the ring (*Figure 1C,D*). The structure suggests that this displacement may be a direct result of Lis1's binding to dynein: the linker position in the dynein alone structure is sterically incompatible with the presence of Lis1 (*Figure 1D*). This is consistent with a model where Lis1 regulates dynein motility through a steric mechanism, by physically blocking the linker's normal position in the no nucleotide state.

We next sought to test if Lis1 sterically blocks the linker in other nucleotide states. As the main mechanical element of dynein, the linker is thought to adopt at least two additional conformations during the ATPase cycle. In the presence of ATP (or ATP plus $V_i$, which leads to the formation of the transition state analog ADP.$V_i$), the linker is displaced across the ring to a position near AAA2 (*Kon et al., 2005*; *Roberts et al., 2009*, *2012*) (the 'pre-powerstroke' position). In the presence of ADP the linker lies over AAA4 in the crystal structure of the *Dictyostelium discoideum* dynein (*Kon et al., 2012*), a 'post-powerstroke' position slightly different from that seen in the *S. cerevisiae* dynein crystal structure in the absence of nucleotide, where the linker is docked onto AAA5 (*Schmidt et al., 2012*). However, since dynein from the same species had not been visualized in both the no nucleotide and ADP states, and because different constructs were used in the studies cited above, it was uncertain whether the AAA4 and AAA5 linker positions corresponded to distinct mechanochemical states or were due to differences between dynein species and/or constructs.

To address this, we first obtained the structure of *S. cerevisiae* dynein alone in the presence of ADP. Conformational sorting revealed that the linker adopts two positions in this condition (*Figure 3A*). One is over AAA4, coinciding with that observed in the *D. discoideum* crystal structure. The other is the AAA5-docked position seen in *S. cerevisiae* dynein with no nucleotide. The AAA4 position was seen only in the presence of ADP and was not detectable in no nucleotide conditions. These results suggest that the linker docks at AAA5 in the absence of nucleotide but can coexist in the AAA4- and AAA5-interacting states in the presence of ADP. Importantly, both the AAA4 linker position in the ADP state and the AAA5 position in the no nucleotide state are sterically incompatible with the presence of Lis1 (*Figure 3B*). Thus, we conclude that binding of Lis1 to the dynein ring results in a displaced linker, away from its normal docking sites under both no nucleotide and ADP conditions.

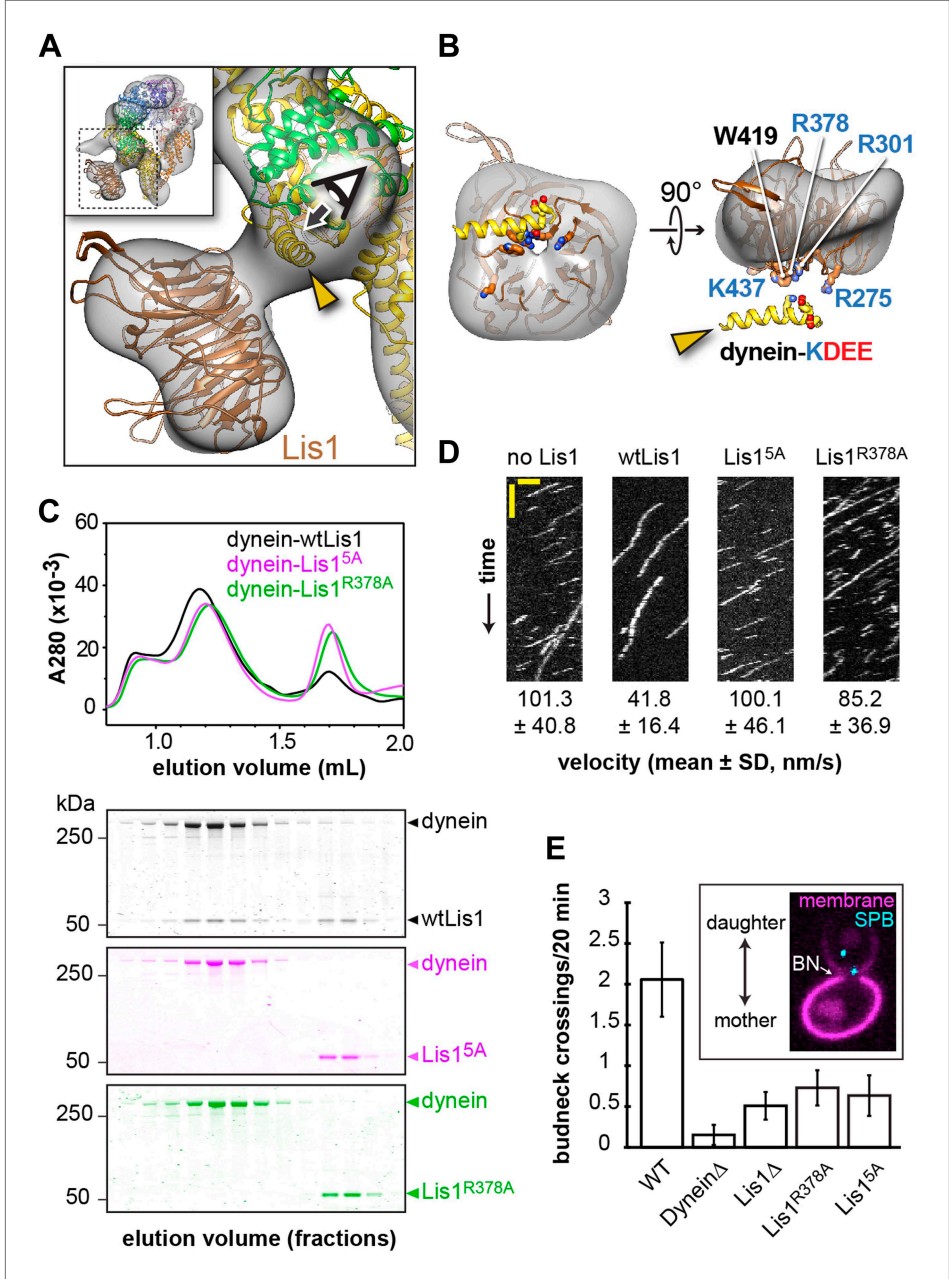

**Figure 2**. Disrupting the putative dynein–Lis1 interface impairs Lis1's ability to bind to and regulate dynein. (**A**) The Lis1 β-propeller engages dynein primarily at a surface helix connecting AAA3 and AAA4 (yellow arrowhead, see *Video 3*). Inset: a zoomed out view. (**B**) (Left) View along the axis highlighted in (**A**) by the eye/arrow; (right) rotated view. Except for the helix (yellow), the dynein density was removed for clarity. Five conserved residues on Lis1 that were mutated to alanine, either in combination (Lis1^5A) or individually, are labeled and shown in atomic representation. Also displayed are residues (KDEE) in dynein known to be involved in the interaction with Lis1 (*Huang et al., 2012*). Basic and acidic residues are labeled in blue and red, respectively. (**C**) No co-migration of dynein and Lis1 was detected by size-exclusion chromatography with the Lis1^5A and Lis1^R378A mutants. Traces show elution profiles of GST-dynein_{331kDa} ('dynein') with wild-type Lis1 (black), Lis1^5A (purple) and Lis1^R378A (green). SDS-PAGE for collected fractions are shown below. (**D**) Kymographs of in vitro motility experiments with TMR-labeled GST-dynein_{331kDa} alone or in the presence of 200 nM wild-type or mutant Lis1. Horizontal scale bar = 2 μm, vertical = 20 s, N = 274–542. (**E**) In vivo spindle oscillation assays comparing *S. cerevisiae* strains carrying either wild-type or mutant Lis1 or full deletions of dynein or Lis1. Inset is a Z-projection of a dividing cell with markers for the membrane (purple)

*Figure 2. Continued on next page*

*Figure 2. Continued*

and spindle pole bodies (SPBs) (cyan). BN = bud neck. Bud neck crossings by the SPBs were counted over 20 min. WT N = 53, DyneinΔ N = 32, Lis1Δ N = 55, Lis1$^{R378A}$ N = 58, Lis1$^{5A}$ N = 47. For each strain the mean and SE are shown.

The following figure supplements are available for figure 2:

**Figure supplement 1**. Probing of the proposed dynein–Lis1 interface by mutagenesis.

**Figure supplement 2**. Velocity distributions for dynein alone or in the presence of wild-type or mutant Lis1.

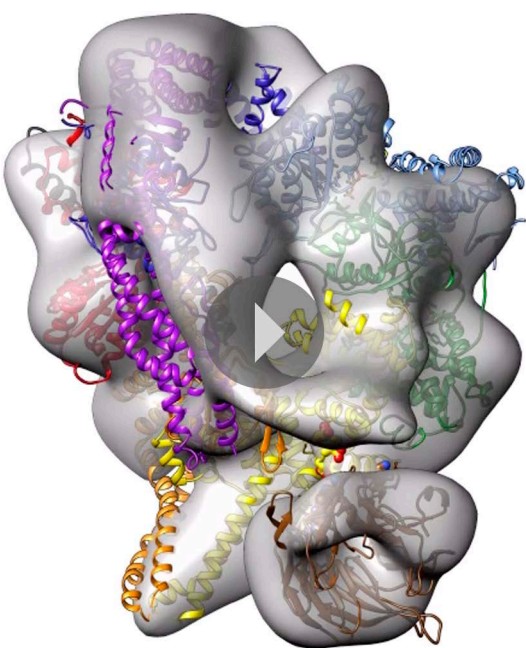

**Video 3**. The dynein–Lis1 interface. The movie shows the 3D reconstruction of dynein–Lis1, with the crystal structure of the dynein motor domain and the Lis1 homology model docked in. After a few frames, the EM density disappears to show only the atomic structures and the view changes to show the interaction between dynein and Lis1 in closer detail, finishing with an open-book view of Lis1. The conserved residues that were mutated in Lis1 are annotated as well as the conserved residues in the AAA4 helix in dynein that have been shown to be necessary for Lis1 binding. Note: the rotational fit of the Lis1 propeller within the Lis1 EM density is uncertain at the current resolution of the dynein–Lis1 map.

We then used two approaches to determine if Lis1 also influences the position of the linker at its pre-powerstroke position, near AAA2. First, we designed a monomeric dynein construct to use fluorescence resonance energy transfer (FRET) to measure linker movement to AAA2. We based our design, which used *S. cerevisiae* dynein, on a linker sensor developed for *D. discoideum* dynein (*Kon et al., 2005*). In our construct, we fused an eGFP donor to the N-terminus of the linker and coupled a tetramethylrhodamine (TMR) acceptor to AAA2 via a small acetyl-CoA-binding tag (ybbR [*Yin et al., 2005*]) (*Figure 3C*). This dynein construct (GFP-dynein$_{FRET/A2}$) slides microtubules robustly, with gliding rates ~90% of wild-type dynein (GFP-dynein$_{331kDa}$) (*Figure 3—figure supplement 1A,B*), showing that the tags are compatible with motor function. Under no nucleotide conditions, and in the absence of Lis1, this construct showed a low FRET efficiency (~2%), as expected when the linker is docked at AAA5 (post-powerstroke position) and the FRET donor and acceptor are far apart (*Figure 3D*). In the presence of ATP and vanadate (V$_i$), which trap dynein as an ADP.V$_i$-bound complex after hydrolysis, FRET increased to ~26% (*Figure 3D*). Under these conditions, the linker is biased towards the pre-powerstroke position at AAA2 and the fluorophores lie closer together. When Lis1 was added to ADP.V$_i$–dynein, there was no significant change in the FRET efficiency relative to ADP.V$_i$–dynein alone (*Figure 3D*). When dynein was incubated with Lis1 before adding ATP and V$_i$, the FRET efficiency decreased somewhat but remained close to that observed for ADP.V$_i$–dynein alone (*Figure 3D*). These results suggest that Lis1 does not affect the linker pre-powerstroke AAA2 position (ATP + V$_i$ added before Lis1) and has only a minor effect on linkers undergoing the AAA5 to AAA2 transition (Lis1 added before ATP + V$_i$).

As a second method to determine if Lis1 affects the linker's ability to reach the AAA2 position under ADP.V$_i$ conditions, we determined the EM structure of the ADP.V$_i$–dynein–Lis1 complex (Lis1 added before ATP + V$_i$). We could resolve the linker in the expected pre-powerstroke position towards AAA2 and the density for the Lis1 β-propeller at the AAA3/4 junction (*Figure 3E,F*). In summary, these results indicate that the presence of Lis1 does not interfere with linker movement towards its

**Table 2.** Dynein:Lis1 ratios in complexes purified by size-exclusion chromatography

| | GST-dynein$_{331kDa}$ | Lis1 | Lis1 (normalized to WT ratio) |
|---|---|---|---|
| WT Lis1 | 0.82 ± 0.01 | 0.18 ± 0.01 | 1.00 |
| Lis1$^{R275A}$ | 0.85 ± 0.01 | 0.15 ± 0.01 | 0.80 |
| Lis1$^{R301A}$ | 0.88 ± 0.01 | 0.12 ± 0.01 | 0.62 |
| Lis1$^{R378A}$ | 1.00 ± 0.00 | 0.00 ± 0.00 | 0.00 |
| Lis1$^{W419A}$ | 1.00 ± 0.00 | 0.00 ± 0.00 | 0.00 |
| Lis1$^{K437A}$ | 0.85 ± 0.01 | 0.15 ± 0.01 | 0.80 |
| Lis1$^{5A}$ | 1.00 ± 0.00 | 0.00 ± 0.00 | 0.00 |

In relation to **Figure 2** and **Figure 2—figure supplement 1**. Fractions were run on SDS-PAGE, stained with SYPRO red, and the bands corresponding to GST-dynein$_{331kDa}$ and wild-type/mutant Lis1 were quantified using ImageJ. The quantification was done using three adjacent lanes corresponding to the peak from size-exclusion. Values are averages of the three lanes ± SD. The ratio for each mutant normalized against that of wild-type Lis1 is also shown.

pre-powerstroke position at AAA2. In contrast, Lis1 occludes the linker binding sites at its two post-powerstroke positions (AAA4 and AAA5).

## ATP turnover in the presence of Lis1 requires a hydrolysis-competent AAA1 and a functional AAA5 linker-docking site

We and others have shown that Lis1 reduces dynein's velocity without significantly affecting the motor's overall ATPase rate (**Yamada et al., 2008**; **McKenney et al., 2010**; **Huang et al., 2012**). However, which of dynein's AAA+ modules is responsible for this continued ATP hydrolysis was not previously addressed. To determine this, we measured microtubule-stimulated ATPase rates, with and without Lis1, in different monomeric dynein constructs.

As expected, dynein monomers continued to hydrolyze ATP in the presence of Lis1 at levels similar to those of dynein alone (**Figure 4A**). This hydrolysis, however, was virtually abolished, both in the presence and absence of Lis1, in a construct where AAA1, the main site of ATP hydrolysis in dynein (**Gibbons et al., 1987**), was rendered hydrolysis-deficient with an E to Q mutation in its Walker B motif (**Kon et al., 2004**) (**Figure 4B**). This result suggests that an intact ATP hydrolysis site at AAA1 is required for ATPase activity in the presence of Lis1.

Given that Lis1 binds at AAA4, one of the hydrolysis-competent AAA+ modules in dynein, it was possible that Lis1 might be stimulating ATP hydrolysis at that site, with AAA1 playing only an indirect role. However, dynein carrying an E to Q mutation in the Walker B motif of AAA4 (**Cho et al., 2008**) showed a near wild-type ATPase rate with or without Lis1 (**Figure 4C**). Therefore, a hydrolysis-competent AAA4 is not required for the ATPase activity observed in the presence of Lis1.

Mutations in AAA5 (an AAA+ module that cannot bind ATP) that prevent linker docking have been shown to severely reduce dynein's ATPase activity (**Schmidt et al., 2012**). We wondered whether Lis1 binding might rescue this mutation and restore ATPase activity to dynein. This was not the case; dynein constructs carrying the AAA5 mutation did not hydrolyze ATP even in the presence of Lis1 (**Figure 4D**).

Taken together, these results indicate that sustained ATP hydrolysis in a Lis1-regulated dynein requires a hydrolysis-competent AAA1 and a functional linker-docking site at AAA5.

## Removing Lis1's steric block by shortening dynein's linker makes the motor Lis1-insensitive

The experiments discussed above showed that Lis1 does not regulate dynein by affecting the linker's ability to reach its pre-powerstroke position at AAA2. Our dynein–Lis1 structure shows that Lis1, however, does affect post-powerstroke linker positions as Lis1 and the linker are sterically incompatible in no nucleotide and ADP conditions (**Figure 1D,E**, **Figure 3B**). We wondered whether motility regulation was a result of this steric blocking by Lis1. Specifically, we wanted to test the hypothesis that steric blocking of the linker is necessary for inducing dynein's Lis1-dependent state of persistent microtubule attachment. To test this hypothesis, we used a dynein construct with a truncated linker that is long enough to form a functional motor but is too short to be sterically blocked by Lis1. This construct is generated by deleting 145 amino acids at the N-terminus of the dynein motor (**Figure 5A**).

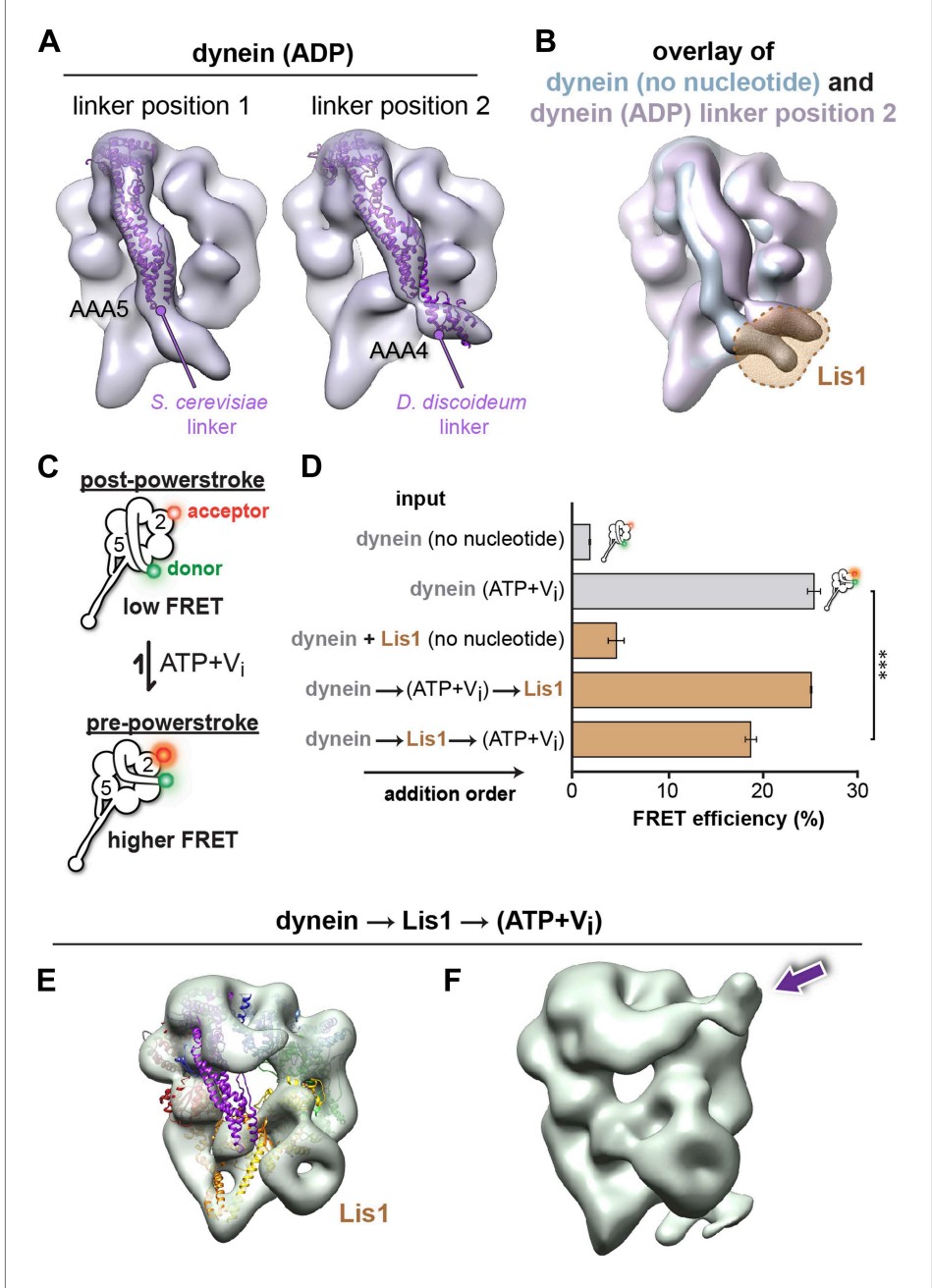

**Figure 3**. Lis1 sterically blocks the linker domain's normal position on dynein's ring in ADP and no nucleotide conditions but does not prevent it from reaching the pre-powerstroke position at AAA2. (**A**) Cryo-NS maps of *S. cerevisiae* dynein in 100 μM ADP displaying the linker next to either AAA5 (left) or AAA4 (right). The *S. cerevisiae* linker domain (lacking nucleotide at AAA1, PDB ID: 4AKG [*Schmidt et al., 2012*]) and the *D. discoideum* linker domain (with ADP at AAA1, PDB ID: 3VKG [*Kon et al., 2012*]) are displayed in purple ribbon representation and have been docked into the linker-AAA5 and linker-AAA4 maps, respectively. To enable unambiguous comparison of linker positions between the EM density and crystal structure, we aligned each EM map to the corresponding dynein motor domain crystal structure after computationally removing the linker. (**B**) The dynein maps in no nucleotide (blue) and ADP (purple) conditions (the latter with the linker at the AAA4 location) are overlaid to compare linker positions. The location of Lis1 in the dynein–Lis1 map is shown as a transparent brown density. Both linker positions are sterically incompatible with the presence of Lis1. Note: since the ADP AAA5 linker position is the same as that seen under no nucleotide conditions, we only show the ADP map with the linker at AAA4. (**C**) Schematic representation of the dynein FRET construct used to test dynein's linker swing in the presence

*Figure 3. Continued on next page*

*Figure 3. Continued*

of Lis1. eGFP (green sphere–donor) was fused to the N-terminus of the linker domain, and TMR (red sphere–acceptor) was inserted into the AAA2 domain in the ring. A pre-powerstroke linker position, where the linker moves close to AAA2 in ATP plus vanadate ($V_i$) conditions, would display an increased FRET efficiency between the two fluorophores (bottom) relative to the no nucleotide state, where the linker is docked at AAA5 (top). (**D**) FRET efficiency between the eGFP and TMR fluorophores in the absence or presence of 200 µM ATP + $V_i$ and 840 nM Lis1, \*\*\*p < 0.001. The order of addition for the reactions containing ATP + $V_i$ and Lis1 is indicated by arrows. Averages of three experiments ± SD are shown. (**E**) Cryo-NS reconstruction of dynein–Lis1 in ATP + $V_i$ conditions with the crystal structure of the motor domain docked in (PDB ID: 4AKG [*Schmidt et al., 2012*]). The Lis1 density is indicated. (**F**) At lower contour levels, the N-terminal portion of the linker can be resolved (purple arrow).

The following figure supplement is available for figure 3:

**Figure supplement 1**. FRET analysis of linker movement towards the pre-powerstroke position in the presence of Lis1.

We first verified this construct functionally and structurally. A dimeric dynein motor containing this shortened linker shows robust motility properties in in vitro motility assays (*Figure 5—figure supplement 1A,B*) (*Reck-Peterson et al., 2006*). We also tested whether shortening the linker affects the microtubule-stimulated ATPase activity of dynein monomers. Monomers with a short linker showed ATPase levels comparable to those seen with a full-length linker, both in the context of a wild-type set of AAA+ modules and in the linker docking-deficient AAA5 mutant (*Figure 5—figure supplement 1D*). As a monomer, the short linker construct can bind Lis1 as shown both by their co-migration in size-exclusion chromatography (*Figure 5—figure supplement 1C*) and by our ability to obtain a 3D reconstruction of the short linker dynein–Lis1 complex (*Figure 5B*). Central to our testing the steric block hypothesis, our 3D structure of the short linker dynein–Lis1 complex shows the same conformation for the linker in the presence of Lis1 as we had observed for the full-length linker in the absence of Lis1 (*Figure 1*, *Figure 5A,B*). Therefore, the short linker is functional and able to physically bypass Lis1.

To directly test whether Lis1 was capable of regulating dynein with a short linker, we used a single-molecule microtubule release assay (*Figure 5C*). In this study, the duration of single monomeric dynein's attachments to microtubules can be measured by kymograph analysis in a flow chamber by TIRF microscopy (*Huang et al., 2012*). Addition of ATP triggers a low-affinity state in dynein (*Kon et al., 2005*; *Imamula et al., 2007*; *Huang et al., 2012*) and the dynein monomers release from microtubules, resulting in a loss of fluorescence signal. Microtubule rebinding events are short lived, likely corresponding to single turnovers of ATP. In the presence of a full-length linker, Lis1 converted dynein to a state of persistent microtubule attachment and dynein monomers stayed bound in the presence of ATP for extended periods as previously shown (*Figure 5D–F*) (*Huang et al., 2012*). Strikingly, shortening of dynein's linker eliminated Lis1's ability to induce this persistent microtubule-bound state. We quantified the durations of microtubule attachments after the addition of ATP and found the same short-lived attachments seen with dynein in the absence of Lis1 (*Figure 5D–F*). Thus, Lis1 is not capable of regulating microtubule attachment in the short linker construct. These data support a steric mode of dynein regulation where Lis1 physically blocks the linker.

## Discussion

We previously described Lis1 as a 'clutch' for dynein, based on its ability to uncouple the cycles of ATP hydrolysis, which take place in the motor domain, from the cycles of microtubule binding and release at the microtubule binding domain (*Huang et al., 2012*). One of the functional consequences of the dynein–Lis1 interaction is that Lis1 keeps dynein in a persistent microtubule-bound state. In this study, we have determined six 3D EM structures of dynein and dynein–Lis1 in different nucleotide states. By combining these structures with single molecule motility experiments, we have established that Lis1 regulates dynein's microtubule attachment by sterically blocking its linker domain.

Together, our data suggest the following model of dynein regulation by Lis1 (*Figure 6*). In the current view of dynein's mechanochemical cycle, the motor domain encounters the microtubule with ADP.$P_i$ bound at AAA1, with the linker in a pre-powerstroke position at AAA2 (*Kon et al., 2005*; *Roberts et al., 2009*, *2012*). Strong microtubule binding stimulates $P_i$ release, inducing the linker to swing to AAA4 (*Kon et al., 2012*). Finally, linker docking at AAA5 is thought to promote the release

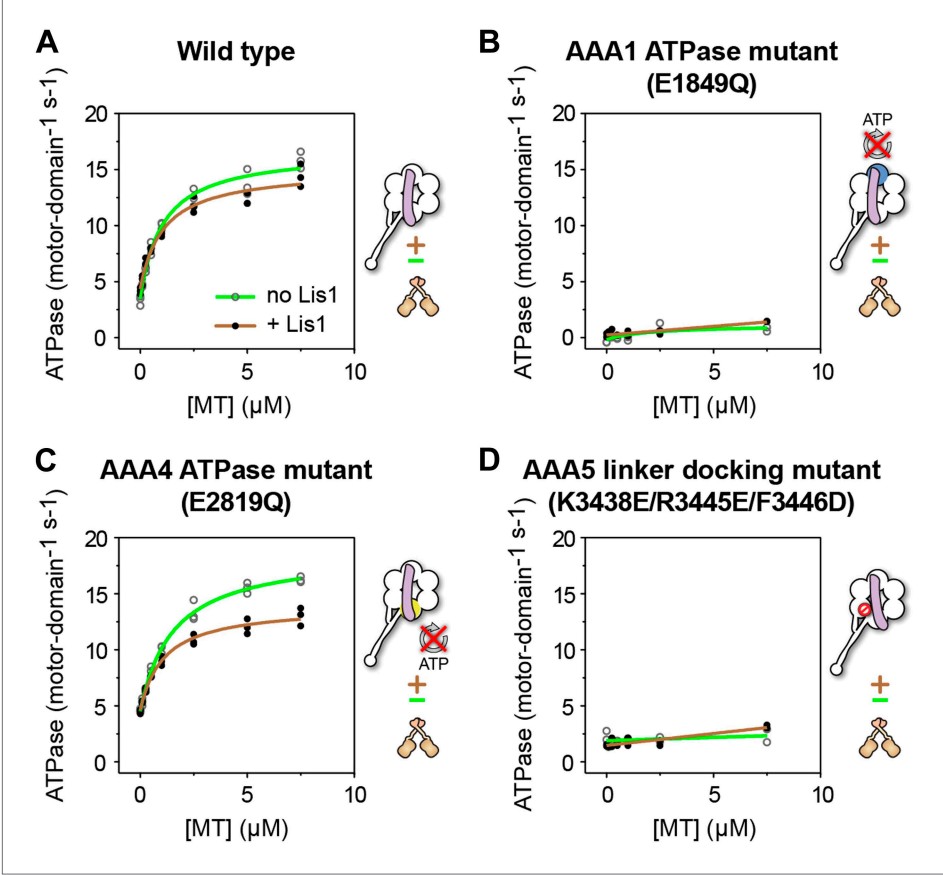

**Figure 4**. ATP turnover in the presence of Lis1 requires a hydrolysis-competent AAA1 and a functional AAA5 linker-docking site. Microtubule-stimulated ATPase activity of dynein monomers carrying (**A**) wild-type AAA+ modules, (**B**) a hydrolysis deficient E1849Q mutation in AAA1 (***Kon et al., 2004***), (**C**) a hydrolysis deficient E2819Q mutation in AAA4 (***Cho et al., 2008***), (**D**) AAA5 mutations (K3438E, R3445E, F3446D) that prevent linker docking (***Schmidt et al., 2012***). ATPase traces are of dynein alone (light green) or in the presence of 140 nM Lis1 (brown). Measurements were done in triplicate (**A** and **C**) or duplicate (**B** and **D**) from one preparation. Diagrams of the dynein constructs used to generate the plots are shown next to them. See ***Table 3*** for fit equation and rate quantifications.

The following figure supplement is available for figure 4:

**Figure supplement 1**. Lis1 binds to dynein ATPase mutants.

of ADP from AAA1, resetting the mechanochemical cycle (***Schmidt et al., 2012***). Our data suggest that when Lis1 is present, the linker retains its ability to adopt the pre-powerstroke AAA2 position but is prevented from reaching its normal post-powerstroke positions at AAA4 and AAA5 on dynein's ring (***Figure 6***). This blocking of the linker by Lis1 is critical for motility regulation; its removal by shortening dynein's linker renders the motor Lis1 insensitive.

Why does Lis1's blocking the linker from adopting its normal post-powerstroke positions prevent dynein's microtubule detachment? One possibility suggested by our structures is that Lis1 disrupts the interaction between the linker and AAA5, preventing normal progression through the mechanochemical cycle. Consistent with this notion, when linker docking at AAA5 is abolished by mutagenesis, dynein displays reduced velocity and prolonged microtubule attachments (***Schmidt et al., 2012***), reminiscent of Lis1's effects. However, while Lis1 has little effect on dynein's ATPase, AAA5 linker docking mutants display severely reduced ATPase rates, both in the absence (***Schmidt et al., 2012***) and the presence of Lis1 (***Figure 4D***, ***Figure 5—figure supplement 1D***). Given these results, it is not clear at this point what the mechanistic basis is for dynein's continuing ATPase in the presence of Lis1. On the one hand, it is possible that the AAA5 mutations may, in addition to preventing linker docking, disrupt

**Table 3.** ATPase assay rate measurements

| Sample | $K_m$(MT)(iM) | $k_{basal}$(Motor domain$^{-1}$.s$^{-1}$) | $k_{cat}$(Motor domain$^{-1}$.s$^{-1}$) |
|---|---|---|---|
| Full-length linker | 1.06 ± 0.16 | 3.51 ± 0.31 | 16.75 ± 0.49 |
| +Lis1 | 1.09 ± 0.20 | 4.36 ± 0.30 | 15.06 ± 0.49 |
| Short linker | 0.92 ± 0.10 | 4.45 ± 0.22 | 16.98 ± 0.32 |
| +Lis1 | 2.05 ± 0.44 | 7.14 ± 0.21 | 16.12 ± 0.61 |
| Full-length linker, AAA4 ATPase mutant (E2819Q) | 1.55 ± 0.14 | 4.53 ± 0.17 | 18.80 ± 0.38 |
| +Lis1 | 1.10 ± 0.15 | 4.60 ± 0.19 | 13.93 ± 0.31 |

Data were fit to the following equation: $k_{obs} = (k_{cat} − k_{basal}) − [MT]/(K_m(MT) + [MT]) + k_{basal}$. $K_m$(MT) is the microtubule concentration that gives half-maximal activation. Values are the averages of triplicate readings ± SE of the fit.

dynein's mechanochemical cycle and thus also prevent ATP hydrolysis. A method to reversibly block linker docking at AAA5 (e.g., via a small molecule) would be required to determine if AAA5 docking is truly required for dynein ATPase activity. On the other hand, Lis1 may uncouple ATP hydrolysis from linker docking at AAA5 through an allosteric effect on the ring. In this scenario, the linker–AAA5 interaction, which is blocked by Lis1, would be required for the conformational changes that ultimately shift dynein's microtubule-binding domain to its low-affinity state, but not for dynein's continuing ATPase activity. Higher resolution structures of the dynein–Lis1 complex will be required to establish whether Lis1 has an effect on the structure of dynein's ring.

It is conceivable that blocking of the normal linker-docking sites by Lis1 might induce a new interaction between the linker and the AAA+ ring. Similarly, Lis1 may interact specifically with the linker itself. Either (or both) of these scenarios could in turn be responsible for preventing microtubule release. However, current evidence does not favor these possibilities. Low sequence conservation in the portion of the linker facing Lis1 argues against a specific Lis1–linker interaction (*Figure 1—figure supplement 1I*). Likewise, a specific interaction between a Lis1-displaced linker and dynein's ring is not supported by the apparent conformational heterogeneity of the N-terminus of the linker in the presence of Lis1, where 3D sorting is required to resolve linker positions (*Figure 1—figure supplement 1F–H*). Also mutating five amino acids on AAA4, proximal to the linker's displaced position (the most likely candidates to interact with the displaced linker), had minimal effect on Lis1-mediated motility regulation (*Figure 1—figure supplement 2*). A direct test of whether specific interactions exist among these different elements will also require a higher resolution structure, where the rotational orientation of the Lis1 homology model within its density in the EM map is unequivocal and specific interactions between the linker and Lis1 as well as the linker and the dynein ring can be distinguished from physical proximity.

In conclusion, our data show that Lis1, a conserved dynein regulator, directly disrupts dynein's mechanochemical cycle by physically blocking conformations that are required to couple the cycles of ATP hydrolysis taking place in the motor domain from those of track binding and release happening at the microtubule binding domain. This allows Lis1 to keep dynein in a persistent microtubule-bound state. This modulation of dynein's interaction with its microtubule track likely contributes to dynein's ability to carry out the variety of cellular functions it performs in different organisms, given the conservation of the amino acids at the dynein–Lis1 interface. For example, Lis1 is involved in initiation of cargo transport (*Lenz et al., 2006*; *Egan et al., 2012*; *Moughamian et al., 2013*), in transport of high load cargo (*McKenney et al., 2010*), and in targeting dynein molecules to the cell cortex via the microtubule plus end (*Lee et al., 2003*; *Sheeman et al., 2003*; *Roberts et al., 2014*). The displaced linker observed in the presence of Lis1 in our 3D dynein–Lis1 reconstruction may contribute to this latter task, generating an 'unmasked' tail domain that has been shown necessary for cortical dynein localization (*Markus and Lee, 2011*). In the case of the mammalian proteins, dynein and Lis1 were previously shown to form a stable complex only in ATP and $V_i$ conditions (*McKenney et al., 2010*). Our 3D reconstruction of dynein–Lis1 under those conditions suggests that this might be a consequence of the linker's moving to its pre-powerstroke site at AAA2, where the linker and Lis1 are no longer sterically incompatible.

The work presented here has helped dissect the molecular mechanism by which Lis1 regulates a single dynein motor domain. The next challenge will be to understand the interactions between Lis1

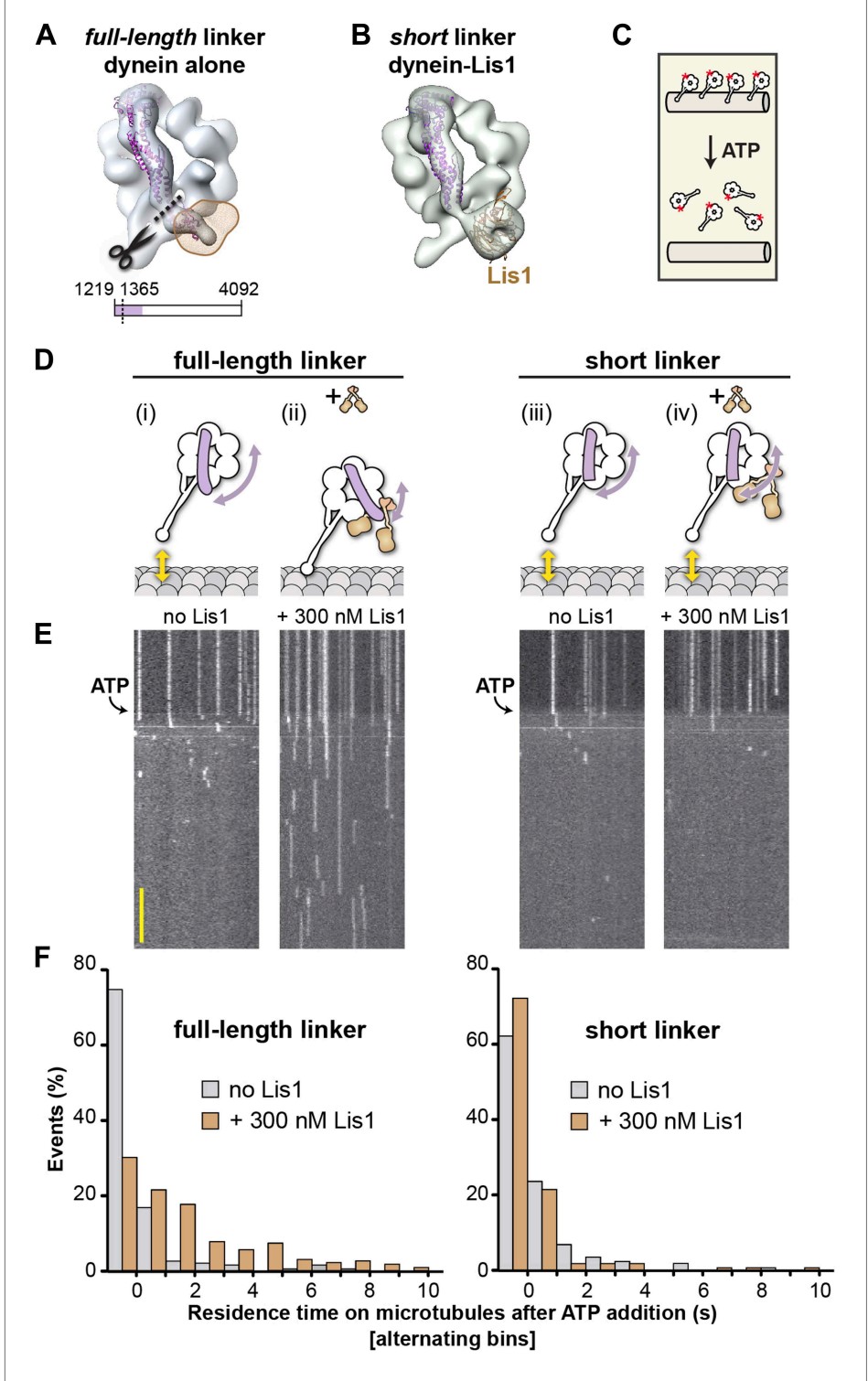

**Figure 5**. A shortened linker that can physically bypass Lis1 renders dynein Lis1 insensitive. (**A**) A short linker construct was designed by docking the crystal structure of the *D. discoideum* linker (purple ribbon) (PDB ID: 3VKG [*Kon et al., 2012*]) into our EM map of dynein alone and overlaying the position of Lis1 (brown mesh). Truncating the linker at residue 1365 (dashed line) yields a linker that is functional (see *Figure 5—figure supplement 1*) but that can no longer contact Lis1. (**B**) Cryo-NS reconstruction of the short linker dynein–Lis1 complex; the linker assumes the same conformation with Lis1 bound as in the absence of Lis1. (**C**) Diagram of the single-molecule

*Figure 5. Continued on next page*

*Figure 5. Continued*

microtubule release assay we used to test Lis1 regulation of dynein. Release from microtubules of TMR-labeled (red asterisk) dynein monomers on addition of ATP is monitored by TIRF microscopy. (**D**) Diagrams of predicted outcomes. Dynein's linker domain in purple, microtubule in gray, Lis1 in brown. (i) Dynein monomers release from microtubules in ATP conditions in the absence of Lis1. (ii) Our model proposes that Lis1 sterically blocks a full-length linker from assuming the normal conformation on dynein's ring, keeping dynein bound to the microtubule. (iii) In the absence of Lis1, shortening the linker would have no effect on dynein's mechanochemical cycle. (iv) Our model predicts that a shortened linker that can bypass the Lis1 steric block should render dynein insensitive to Lis1. (**E**) Kymographs of TMR-labeled full-length (left) or short linker (right) dynein molecules. After pre-binding to microtubules, release of dynein molecules is monitored after addition of 5 mM ATP, with and without 300 nM Lis1. Kymographs correspond to the dynein constructs shown in (**D**). Scale bar = 5 s. (**F**) Quantification of the kymographs in (**D**), showing the duration of microtubule attachment after addition of ATP, in the absence (gray) or presence (brown) of Lis1. Data were binned into 1 s intervals and the histograms show alternating no Lis1 and +Lis1 bars. Rare attachments longer than 10 s were excluded from the analysis and plot, N = 179–183.

The following figure supplement is available for figure 5:

**Figure supplement 1**. The short linker dynein construct shows robust motility, hydrolyzes ATP, and binds Lis1.

and dynein dimers and of those with other regulatory factors. Future structural studies with full-length dimeric dynein–Lis1–Nudel complexes, free and bound to microtubules, will be required to answer these exciting questions.

# Materials and methods

## Yeast strain construction

The *S. cerevisiae* strains used in this study are listed in *Table 1*. Deletions or modifications of endogenous genomic copies of the dynein heavy chain (*DYN1*) and Lis1 (*PAC1*) were done using PCR-based methods as previously described (*Longtine et al., 1998*), using the URA3/5FOA 'pop-in/pop-out' method (*Guthrie and Fink, 1991*). Transformations were performed using the standard lithium acetate method (*Gietz and Woods, 2002*). Point mutants were generated using the PCR stitching method and verified by DNA sequencing.

## Protein expression and purification

Cultures of *S. cerevisiae* for protein purification were grown, harvested, and frozen as described previously (*Reck-Peterson et al., 2006*). Dynein and Lis1 constructs were purified and labeled as described previously (*Reck-Peterson et al., 2006*; *Huang et al., 2012*), except that a modified TEV buffer for Lis1 purification was used; 50 mM Tris–HCl (pH 8.0), 150 mM potassium acetate, 2 mM magnesium acetate, 1 mM EGTA, 5% glycerol, 1 mM DTT, and 1 mM PMSF.

## EM sample preparation

We chose to use cryo-NS EM, where a carbon support, combined with a heavy metal stain, resulted in highly reproducible grids with high contrast. Prior attempts at getting dynein reproducibly in open holes for standard cryo-EM were unsuccessful. Furthermore, cryo-EM on continuous carbon gave micrographs where individual dynein particles were difficult to see above the noise. The high reproducibility we were able to achieve with cryo-NS allowed us to sample a much greater range of constructs/nucleotide conditions in the same time frame than we would otherwise have been able to do in unstained, unsupported conditions. Most importantly, the improved contrast was instrumental in allowing us to sort the different dynein conformations that were present in most of our data sets. 4 μl of monomeric dynein (80–120 nM), or monomeric dynein pre-incubated for 10 min with Lis1 dimer at a 1.5-fold excess (120–180 nM), was applied to a glow discharged, continuous carbon coated, C-flat EM grids (Protochips, Raleigh, NC). Dynein samples stated to be prepared in no nucleotide conditions were treated with apyrase (0.14 U/ml) for 15 min prior to grid application to hydrolyze residual ADP left over from the dynein purification procedure. Dynein samples stated to be prepared in ADP and ATP + $V_i$ conditions contained 100 μM ADP and 500 μM Mg-ATP/NaVO$_4$, respectively. For the latter, nucleotide was added after the dynein–Lis1 pre-incubation step. Once applied to the grid, the samples were stained with 2% uranyl formate by floating the grid sample face down on a pool of stain. Samples were then sandwiched with a thin

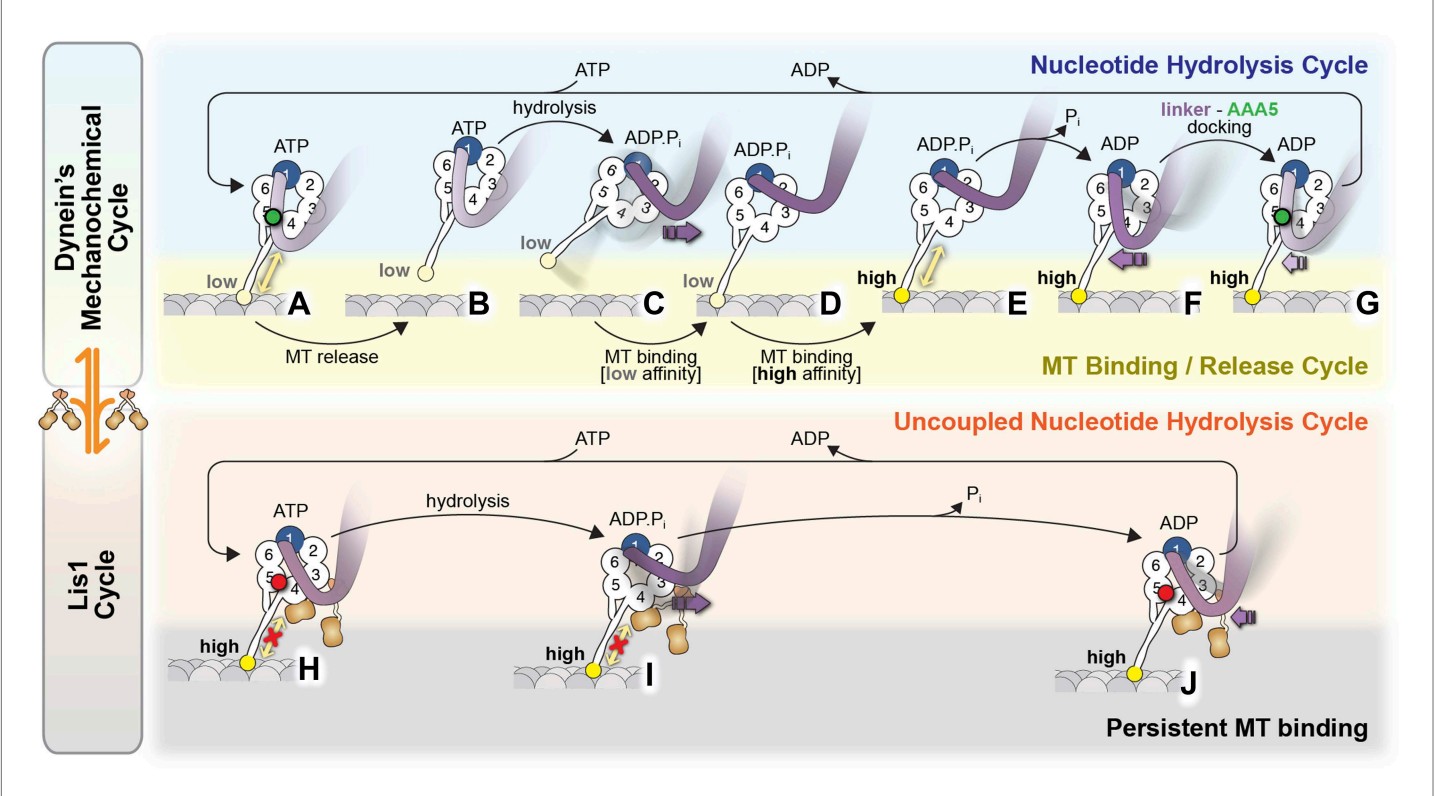

**Figure 6**. Model for the regulation of dynein by Lis1. (**A**–**G**) Current view of dynein's mechanochemical cycle. (**A**) ATP binding to AAA1 induces the low-affinity conformation in dynein's microtubule-binding domain and (**B**) release from the microtubule. (**C**) The linker domain changes its position from AAA5 towards AAA2, the 'pre-powerstroke' and ATP is hydrolyzed. (**D**) Binding of dynein to a new site on the microtubule triggers a change in the microtubule-binding domain to its high affinity conformation (**E**). (**F**) Release of $P_i$ results in the 'powerstroke', a movement of the linker back towards AAA5. (**G**) Docking of the linker at AAA5 is thought to promote nucleotide exchange at AAA1, resetting the motor for a new cycle. (**H**–**J**) Model for the Lis1-regulated cycle. Lis1 prevents the linker from completing its normal conformational cycle, keeping dynein in a persistent microtubule-attached state, despite continuing ATP hydrolysis. (**H**) Binding of Lis1 to dynein blocks the linker from docking onto the ring at AAA5, preventing the conformational changes in the stalk and microtubule binding domain that ultimately result in dynein's release from the microtubule. (**I**) The linker is still capable of moving to the pre-powerstroke position at AAA2 in the presence of Lis1, and ATP is hydrolyzed. (**J**) Presumably, by analogy to the dynein alone cycle, $P_i$ release triggers the power-stroke, but Lis1 sterically blocks the linker's normal position on dynein's ring in the ADP state. Our current understanding of Lis1 regulation does not yet explain the mechanism of nucleotide exchange at AAA1.

layer of freshly evaporated carbon, and grids were lightly blotted from the non-sample containing side and plunged into liquid nitrogen. Grids were then stored at liquid nitrogen temperatures.

## EM data collection

Samples were imaged at liquid nitrogen temperatures using a Gatan 626 cryo holder (Gatan, Inc., Pleasanton, CA) on a Tecnai F20 TEM microscope (FEI, Hillsboro, OR), operating at 120 kV, equipped with a US4000 4k × 4k CCD camera (Gatan). Data were collected either manually or automatically using Leginon (*Carragher et al., 2000*). Dynein alone samples (no nucleotide [strain RPY844] and ADP conditions [strain RPY844]) and dynein–Lis1 (ATP + $V_i$ conditions [strains RPY1302 and RPY816]) were imaged at 62,000× nominal magnification (1.73 Å/pixel). Dynein–Lis1 (no nucleotide [strains RPY1302 and RPY816]) was imaged at 50,000× nominal magnification (2.14 Å/pixel). Short linker dynein–Lis1 (no nucleotide [strains RPY1436 and RPY816]) was imaged at 80,000× nominal magnification (1.34 Å/pixel). Low-dose conditions during imaging (dose ~25 e⁻/Å²) were used for all data sets, and micrographs were collected using a defocus range of −0.6 to −1.5 μm.

## EM image pre-processing

For all data sets, ~1,000 particles were initially selected manually in Boxer (EMAN1) (*Ludtke et al., 1999*) and reference-free 2D classified in IMAGIC (*van Heel et al., 1996*) to give class averages that

were then used as templates for automated particle picking in Appion (*Lander et al., 2009*). Reference-free 2D classification in IMAGIC was subsequently used on the data sets to remove averages with blurred appearance or incorrect size. CTF determination and correction of image phases were carried out in Appion using Ace2 (NRAMM). Particles were band-pass filtered (high-pass = 250 Å, low-pass = 3 × sampling) in Imagic and normalized in Xmipp (*Sorzano et al., 2004*). For 3D classification and initial 3D refinement particles were binned by two; final 3D refinements were carried out using unbinned data.

## EM image processing

### Dynein (no nucleotide)

An initial model was generated using the *S. cerevisiae* dynein motor domain crystal structure (PDB ID: 4AKG [*Schmidt et al., 2012*]), low-pass Fourier filtered to 80 Å. Initial 3D refinement was carried out in EMAN2 (*Tang et al., 2007*). The resulting map was filtered to 40 Å and used as an initial model for 3D classification in RELION (*Scheres, 2012*). Five classes were generated. Particles from four of the classes were combined and refined in RELION against the class 5 map, filtered to 40 Å. 3D refinement converged after 18 iterations. The final map contained 31,839 particles (from 38,463 total) and the 'gold-standard' resolution using an FSC cut-off of 0.143 was 14.8 Å. The final map was filtered according to local resolutions (*Cardone et al., 2013*).

### Dynein (ADP)

The linker domain has been shown to have a different position relative to the dynein ring in ADP conditions in *D. discoideum* dynein compared with that of *S. cerevisiae* dynein in no nucleotide conditions (*Kon et al., 2012*; *Schmidt et al., 2012*). To avoid initial model bias of linker position, the domain was computationally removed from residue 1,620 of PDB file 4AKG (*Schmidt et al., 2012*). The resulting map was filtered to 50 Å and used as a starting model for an initial refinement in SPIDER (*Frank et al., 1996*) to regain linker density at a position derived solely from the data. This map was then filtered to 40 Å and used as an initial model for 3D classification in RELION. Five classes were generated. Four of the classes showed the linker at the no nucleotide position (position 1) and 1 class at the shifted position seen with *D. discoideum* dynein in ADP conditions (position 2). The particles in classes corresponding to each conformation were further refined in RELION and the refinements converged after 13 and 10 iterations for linker position 1 and 2 data sets, respectively. The final maps contained 7,630 and 3,983 particles (from 17,256 total) and the 'gold-standard' resolutions using an FSC cut-off of 0.143 were 18.3 and 19.5 Å for linker position 1 and 2 maps, respectively. The final maps were filtered according to local resolutions (*Cardone et al., 2013*).

### Dynein–Lis1 (no nucleotide)

The same initial model as described for dynein (no nucleotide) above was used to 3D refine an initial data set of dynein–Lis1 (RPY844, RPY816) in EMAN2. This map was then filtered to 60 Å, and EMAN2 was used to refine a larger data set of dynein–Lis1 using a dynein lacking any tags on the end of the linker (RPY1302). The resulting map was filtered to 40 Å and used for 3D classification in RELION. This process was repeated with different requested class numbers. Linker position was observed to vary across classes (*Figure 1—figure supplement 1F–H*). In a run with seven generated classes, classes with most density for the linker (class 1 and 6) were combined and refined against the class 1 map filtered to 40 Å in RELION. Three-dimensional refinement converged after 16 iterations. The final map contained 10,129 particles (from 35,472 total) and the 'gold-standard' resolution using an FSC cut-off of 0.143 was 21.4 Å. The final map was filtered according to local resolutions (*Cardone et al., 2013*).

### Short linker dynein–Lis1 (no nucleotide)

The same initial model as described for dynein (no nucleotide) was filtered to 50 Å and used for initial 3D classification of the data set in RELION. Five classes were generated. Particles from 1 class were further refined against the class map in RELION. The refinement converged after 14 iteractions and contained 11,818 particles (from 34,805 total). The gold-standard resolution using an FSC cut-off of 0.143 was 15.4 Å. The final map was filtered according to local resolutions (*Cardone et al., 2013*).

### Dynein–Lis1 (ATP + V$_i$)

The map of dynein–Lis1 (no nucleotide) filtered to 60 Å was used as a starting model for initial refinement in EMAN2. The resulting map was filtered to 40 Å and used for 3D classification in RELION.

Most classes showed the linker position unresolved, indicative of variability in location as previously observed (*Roberts et al., 2009*, *2012*), but one class resolved the linker near AAA2 when viewed at lower contour levels. The particles in this class were refined against the class map in RELION and refinement converged after 16 iterations. The final map contained 1,072 particles (from 6,600 total), and the 'gold-standard' resolution using an FSC cut-off of 0.143 was 23.1 Å.

## Accession numbers

EM maps have been deposited with the EMDataBank. Accession codes as follows; dynein–Lis1 (no nucleotide conditions) EMDB-6008; dynein alone (no nucleotide conditions) EMDB-6013; dynein alone (ADP conditions) with position 1 and 2 linker domains, EMDB-6015 and EMDB-6014 respectively; dynein–Lis1 (ATP + $V_i$ conditions) EMDB-6016; short linker dynein–Lis1 (no nucleotide conditions) EMDB-6017. For each entry, in addition to the final masked and filtered maps, raw half maps for each reconstruction have been deposited. We also deposited an XML file of the FSC plot between the dynein alone map and the fitted crystal structure of the motor domain (PDB ID: 4AKG [*Schmidt et al., 2012*]), as a supplementary file to the dynein alone submission (EMDB-6013).

## Size-exclusion chromatography

Dynein and Lis1 were tested for complex formation, and Lis1 mutants were tested for structural integrity (*Figure 2—figure supplement 1C*) by size-exclusion chromatography. 400–800 nM dynein and 475–800 nM Lis1 were loaded separately or after being mixed for 10 min at 4°C. Samples were fractionated on a Superose 6 PC 3.2/30 column using an ÄKTAmicro system (GE Healthcare) that had been equilibrated with degassed gel filtration buffer (50 mM Tris–HCl pH 8.0, 150 mM potassium acetate, 2 mM magnesium acetate, 1 mM EGTA, 1 mM DTT). Fractions (50 µl or 90 µl) were analyzed by SDS-PAGE on 4–12% Tris-Bis gels (Invitrogen, Grand Island, NY) with SYPRO Red staining (Invitrogen) and imaged using an ImageQuant 300 (BioRad, Hercules, CA) or Typhoon (Amersham, UK) gel imaging system.

## Single-molecule microscopy

Single-molecule motility assays were performed using flow chambers as previously described (*Case et al., 1997*). Dynein was labeled with TMR (Promega, Madison, WI), and microtubules contained ~10% biotin-tubulin for surface attachment and ~10% HyLite488-tubulin (Cytoskeleton Inc., Denver, CO) for visualization. For assays that included Lis1, dynein was incubated with 200 nM Lis1 for 10 min at 4°C prior to addition to the flow chamber. The imaging buffer consisted of 30 mM HEPES (pH 7.2), 50 mM potassium acetate, 2 mM magnesium acetate, 1 mM EGTA, 10% glycerol, 1 mM DTT, 20 mM taxol, 1.25 mg/ml casein, 1 mM Mg-ATP, and an oxygen scavenger system. Images were recorded every 2 s for 5 or 10 min, and dynein velocities and run-lengths were calculated from kymographs generated in ImageJ (National Institutes of Health).

In vitro motility assays were visualized on either a Zeiss Elyra PS.1 microscope with a 100× 1.46 N.A. oil immersion TIRF objective (Carl Zeiss GmbH, Germany) with an Andor EM-CCD camera or an Olympus IX-81 TIRF microscope with a 100× 1.45 N.A. oil immersion TIRF objective (Olympus, Japan) with a Hamamatsu EM-CCD camera. TMR-labeled dynein and HyLite488-microtubules were excited with 561 nm and 488 nm solid state laser lines, respectively. Images were recorded with a 100 ms exposure using Zen Black (Zeiss) or Metamorph software. Microtubule gliding assays and microtubule binding and release assays were performed as described (*Huang et al., 2012*). Control experiments for the microtubule release assays examined dynein release in buffer lacking ATP (*Figure 5—figure supplement 1E*), where dynein remained bound to microtubules as expected (*Huang et al., 2012*) and with dynein lacking N-terminal tags (*Figure 5—figure supplement 1F,G*), where untagged dynein behaved similar to tagged dynein (*Figure 5E,F*).

## Spindle oscillation assay

To track the dynein-dependent movement of spindle pole bodies (SPBs), we used a strain containing a GFP-labeled SPB marker, *SPC110*, and a tdTomato-labeled cell membrane marker, *HXT1* (kindly provided by Jeff Moore, University of Colorado). Mutations were introduced into the *PAC1* (Lis1) locus in this strain. For control experiments, strains containing deletions of the dynein heavy chain (*DYN1*) and *PAC1* loci were constructed. All strains were PCR verified, and mutations were additionally verified by DNA sequencing.

For image analysis, saturated overnight cultures for each strain were diluted to an OD$_{600}$ of 0.1 in a total volume of 5 ml YPD media. The dilution of cultures was staggered such that the data could be

collected for all strains during a single imaging session. Following dilution, each culture was incubated with rotation at 30°C for 3 hr. Hydroxyurea (HU) was then added to a final concentration of 200 mM, and the culture was incubated for an additional 2 hr with rotation at 30°C. The cells were collected by centrifugation, the media was discarded, and the cells were resuspended in 250 µl of fresh YPD + 200 mM HU. The cells were loaded into an Y04C microfluidic yeast plate (CellASIC, EMD-Millipore, Germany) and introduced into the viewing chamber with the ONIX controller (CellASIC). Imaging was performed at the Nikon Imaging Center at Harvard Medical School. All images were collected with a Yokagawa CSU-X1 spinning disk confocal with Borealis modification, on a Nikon Ti inverted microscope equipped with a Plan 60 × 1.4 N.A. objective and the Perfect Focus System (Nikon Corp., Japan). GFP-labeled SPB and tdTomato-labeled cell membrane fluorescence were excited with the 488 nm and 561 nm lines, respectively, from a LMM-5 solid state laser merge module controlled with an ATOF (Spectral Applied Research Inc., Canada). Images were acquired with a Hamamatsu ORCA-AG CCD controlled with MetaMorph 7.0 software. Images were collected for SPBs as 100 ms exposures, spanning 9 × 500 nm Z-sections (4.5 µm total Z stack) every 30 s for a total of 20 min. Cell membranes were imaged as single central Z-sections at the first and last time point. Membrane image pairs were digitally merged to allow for drift analysis; those cells with visible drift were excluded from analysis.

## Image processing and particle tracking

Maximum intensity projections were calculated for Z-series at each time point for GFP-labeled SPBs. At each time point, SPBs were independently detected in the Z-projection using a wavelet detection algorithm (*Aguet et al., 2013*), and the two spindles were tracked throughout the course of the movie using a nearest neighbor tracking method (unpublished Matlab [Mathworks, Natick, MA] scripts). The location of the bud neck and the mother–daughter orientation were determined using the first tdTomato-labeled cell membrane exposure. The locations of tracked SPBs were used to calculate the number of bud neck crossings.

## FRET

To generate the dynein FRET construct, eGFP (the FRET donor) was inserted at the dynein N-terminus and the acceptor site was inserted after L2241 in AAA2. We used the ybbR tag (GGGTVL**DSLEFIASKLA**GGG [*Yin et al., 2005*]) labeled with TMR-CoA (NEB, Ipswich, MA) as the FRET acceptor. Dynein was incubated with or without Lis1 for one hour on ice, followed by apyrase (6.6 U/ml) or 200 µM ATP.$V_i$ for 2 min at room temperature (RT). For some experiments, dynein was first incubated with 200 µM ATP.$V_i$ for 2 min at RT, followed by Lis1 for 1 hr on ice. Assays were performed in 30 mM HEPES (pH 7.2), 50 mM potassium acetate, 2 mM magnesium acetate, 1 mM EGTA, 1 mM DTT and the final concentrations of dynein and Lis1 were 84 nM and 840 nM, respectively. The sample was excited with 485 nm (eGFP) light, and the emitted light was detected from 505 nm to 650 nm in a SpectraMax M5 fluorimeter (Molecular Devices, Sunnyvale, CA) at RT. In order to normalize across experiments, the samples were also excited with 535 nm (TMR) light and the emitted light was detected from 555 nm to 700 nm. To analyze the FRET data, we first subtracted the fluorescence background from the buffer alone. We then used the emission spectra of dynein-labeled with eGFP and free TMR dye alone to decompose each channel in the experimental spectra. FRET efficiencies (E) were calculated using the method of Clegg (*Clegg, 1995*): $E = \{F^a_{FRET}/F^a_{DIR} − \varepsilon^a(485)/\varepsilon^a(535)\}\varepsilon^a(535)/\varepsilon^d(485)$, where the superscripts 'd' and 'a' refer to the donor (eGFP) and the acceptor (TMR), respectively. $F^a_{FRET}$ is the fluorescence intensity of the acceptor excited at 485 nm and $F^a_{DIR}$ is the fluorescence intensity of the acceptor excited at 535 nm. $\varepsilon^d(485)$ $\varepsilon^a(485)$ and $\varepsilon^a(535)$ are the molar extinction coefficients at the designated wavelengths. In our experiments $\varepsilon^a(535)/\varepsilon^d(485) = 37{,}900$ M$^{-1}$ cm$^{-1}$/40,000 M$^{-1}$ cm$^{-1}$ and $\varepsilon^a(485)/\varepsilon^a(535) = 0.2$.

## ATPase assays

Dynein constructs used in ATPase assays were tested for complex formation with Lis1 by size-exclusion chromatography (*Figure 4—figure supplement 1*). ATPase assays were performed using an EnzChek phosphatase kit (Molecular Probes, Thermo Fisher Scientific Inc., Cambridge, MA) as previously described (*Reck-Peterson et al., 2006*; *Cho et al., 2008*). The final reaction consisted of 10–20 nM dynein (monomeric constructs, see *Figure 4* and *Figure 5—figure supplement 1D*), 0 or 140 nM Lis1, 0–7.5 µM taxol-stabilized microtubules, 2 mM Mg-ATP, 200 mM MESG (2-amino-6-mercapto-7-methyl-purine riboside), 1 U/ml purine nucleoside phosphorylase, and assay buffer (30 mM HEPES (pH 7.2), 50 mM potassium acetate, 2 mM magnesium acetate, 1 mM EGTA, 1 mM DTT, and 10 mM taxol).

A SpectraMax384 plate reader (Molecular Devices) was used to monitor the coupled reaction at $OD_{360}$ every 12 s for 10 min. Data were fit according to *Nishiura et al. (2004)*.

## Acknowledgements
We would like to thank Jeff Moore (University of Colorado School of Medicine) for yeast strains, Vu Nguyen, Rogelio Hernandez-Lopez, Michael Cianfrocco, John Srouji, and Julie Huang for their contributions during this work, Morgan deSantis for comments on the manuscript, and Liza Sholl for administrative support. EM data were collected at Harvard's Center for Nanoscale Systems, a member of the National Nanotechnology Infrastructure Network and supported by the National Science Foundation (NSF) (ECS-0335765). Computation was performed in part on the Odyssey cluster supported by Harvard's FAS Science Division Research Computing Group. Single molecule data were collected in part at Harvard's Center for Biological Imaging, supported by an SIG award (RR1S1027990) from the NIH. Spinning disc confocal microscopy data were collected at the Nikon Imaging Center at Harvard Medical School. KT was supported by a Charles King postdoctoral fellowship. AJR is a Sir Henry Fellow supported by the Wellcome Trust (092436/Z/10/Z), sponsored by Peter J Knight, and Stan A Burgess (University of Leeds). SLR-P was supported by the Rita Allen Foundation. SLR-P and AEL were supported by the NIH (R01 GM 107214).

## Additional information

### Funding

| Funder | Grant reference number | Author |
|---|---|---|
| Charles A. King Trust | Postdoctoral Research Fellowship | Katerina Toropova |
| Wellcome Trust | 092436/Z/10/Z | Anthony J Roberts |
| Rita Allen Foundation | | Samara L Reck-Peterson |
| National Institutes of Health | R01 GM 107214 | Samara L Reck-Peterson, Andres E Leschziner |

The funders had no role in study design, data collection and interpretation, or the decision to submit the work for publication.

### Author contributions
KT, SZ, AJR, WBR, Conception and design, Acquisition of data, Analysis and interpretation of data, Drafting or revising the article; BSG, Analysis and interpretation of data, Drafting or revising the article; SLR-P, AEL, Conception and design, Analysis and interpretation of data, Drafting or revising the article

## Additional files

### Major dataset

The following datasets were generated:

| Author(s) | Year | Dataset title | Dataset ID and/or URL | Database, license, and accessibility information |
|---|---|---|---|---|
| Toropova K, Zou S, Roberts AJ, Redwine WB, Goodman BS, Reck-Peterson SL, Leschziner AE | 2014 | Dynein-Lis1 (no nucleotide conditions) EM structure | EMD-6008; http://www.ebi.ac.uk/pdbe/entry/EMD-6008 | Available at the EMDataBank repository (http://www.emdatabank.org/). |
| Toropova K, Zou S, Roberts AJ, Redwine WB, Goodman BS, Reck-Peterson SL, Leschziner AE | 2014 | Dynein (no nucleotide conditions) EM structure | EMD-6013; http://www.ebi.ac.uk/pdbe/entry/EMD-6013 | Available at the EMDataBank repository (http://www.emdatabank.org/). |

| Toropova K, Zou S, Roberts AJ, Redwine WB, Goodman BS, Reck-Peterson SL, Leschziner AE | 2014 | Dynein (ADP conditions), linker position 1 EM structure | EMD-6015; http://www.ebi.ac.uk/pdbe/entry/EMD-6015 | Available at the EMDataBank repository (http://www.emdatabank.org/). |
|---|---|---|---|---|
| Toropova K, Zou S, Roberts AJ, Redwine WB, Goodman BS, Reck-Peterson SL, Leschziner AE | 2014 | Dynein (ADP conditions), linker position 2 EM structure | EMD-6014; http://www.ebi.ac.uk/pdbe/entry/EMD-6014 | Available at the EMDataBank repository (http://www.emdatabank.org/). |
| Toropova K, Zou S, Roberts AJ, Redwine WB, Goodman BS, Reck-Peterson SL, Leschziner AE | 2014 | Dynein-Lis1 (ATP + Vi conditions) EM structure | EMD-6016; http://www.ebi.ac.uk/pdbe/entry/EMD-6016 | Available at the EMDataBank repository (http://www.emdatabank.org/). |
| Toropova K, Zou S, Roberts AJ, Redwine WB, Goodman BS, Reck-Peterson SL, Leschziner AE | 2014 | Short linker dynein-Lis1 (no nucleotide conditions) EM structure | EMD-6017; http://www.ebi.ac.uk/pdbe/entry/EMD-6017 | Available at the EMDataBank repository (http://www.emdatabank.org/). |

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
