## [Decision Letter]

Thank you for sending your work entitled “Lis1 regulates dynein by sterically blocking its mechanochemical cycle” for consideration at *eLife.* Your article has been favorably evaluated by John Kuriyan (Senior editor) and 3 reviewers, one of whom, Gabriel Lander, has agreed to reveal his identity.

The Senior editor and the three reviewers discussed their comments before we reached this decision, and the Senior editor has assembled the following comments to help you prepare a revised submission.

In the present work, the authors use electron microscopy to determine the three-dimensional structure of dynein bound to Lis1, enabling unambiguous identification of this regulator's binding site near the stalk of the motor. Although the resolutions of the reconstructions are moderate (in the 15-20Å range), the resolvable structural details were more than adequate to identify the key interactions that were involved in Lis1 regulation of dynein.

This group had a breakthrough paper in 2012, determining the Lis1-binding domain on the dynein motor domain, and using elegant single molecule studies to develop a clutch model for Lis1-mediated regulation of dynein motility. This work is a logical follow-up, with 3-D structures that support their previous binding data, and a clever short-linker mutant to test their more refined steric-blocking model.

Comparing the structure of the dynein head in the absence and presence of Lis1, the authors noticed that Lis1 binding induces a significant displacement of the dynein linker arm, a critical component of the mechanochemical cycle. Based on this finding, the authors carried out a series of well-designed biochemical and biophysical experiments to probe the mechanical relationship between bound Lis1 and the linker arm. By docking in previously determined atomic coordinates, the resulting pseudo-atomic models served as a framework for the mutagenesis and FRET experiments, which provided insight into the relationship between Lis1, nucleotide state, and the linker arm position. Truncation of the linker arm, single molecule motility experiments, and in vivo assays were performed to confirm the author's model for Lis1 regulation. Ultimately, the authors conclude that Lis1 serves as a molecular barricade, sterically preventing ATP hydrolysis from being converted into the conformational rearrangements that weaken microtubule binding.

Some of the important insights into how Lis1 works that emerge from this work are as follows:

1) That Lis1 occludes to normal position of the dynein linker in the ADP and nucleotide free (APO) conformations of dynein. In the presence of Lis1, the linker is deflected from its normal position. This EM work also provides a nice confirmation that the two conformations observed in yeast(APO) vs Dictyostellium(ADP) crystal structures can both exist in one (yeast) dynein.

2) That the motor can still enter the pre-powerstroke conformation when Lis-1 is bound. This was demonstrated by both an EM structure in the presence of ATP.vanadate and a FRET assay.

3) That the ability of Lis-1 to block release of dynein from microtubules is abolished when the linker is shortened (from the 331kD to 314kD construct). This shows that the N-terminal part of the linker is required for Lis1 function.

The EM work is expertly done, particularly in exploring the conformational variability that exists within a set of particles. The dynein motor is relatively small for detailed structural analyses by EM, and the authors' ability to distinguish between conformational subtleties in the motor arm is an impressive example of their expertise.

In general, the consensus among the reviewers is that this work is an important step forward in understanding dynein motor regulation, and provides a direct view of the conformational rearrangements that are necessary for dynein movement along the microtubule surface. The reviewers have, however, raised a number of points that you should address in preparing a revised manuscript.

1) There is some concern that much of this work follows directly from the outstanding [23] paper published by this group. In this paper, the authors further refine the model developed there, and while the three-dimensional structure adds knowledge, it is limited in detail because the docking of Lis1 is not fully resolved in terms of rotational alignment. The couple of new mutants examined here could also have been predicted based on the previous structure.

The most novel idea here is that the binding of Lis1 provides a steric block to the linker swing. This is tested with a single construct with a truncated linker, and the results support their model. But there is concern that this is a single approach; the paper would have more impact if the authors could demonstrate this key point in a complementary way. The reviewers recognize that this might be a tall order, but it would be good if the authors could respond to this issue.

2) The authors interpret their data with a model in which the linker needs to get to AAA5 in order for the cycle of ATP hydrolysis to continue and so blocking access to AAA5 prevents ATP binding and therefore release from microtubules (which is thought to occur on ATP hydrolysis). The problem with this model, as the authors discuss, is that it would predict that Lis1 would inhibit the dynein ATPase. The data, however, are clear that this is not the case. It is suggested that the authors reduce the emphasis on the model that Lis1 blocks the linker from getting to AAA5. Instead the data could be presented and then this discussed as an option.

More specific points to consider:

Specific issues raised by the reviewers are given below. The authors should use their judgement in how to respond to these points, but if they choose to not address any particular issue they should explain their reasons in the cover letter accompanying the resubmitted manuscript.

1) The authors employed cryo-negative staining for their EM analysis. This involved and tedious technique is not commonly used by EM labs, as it is difficult to master. Surely the authors had a reason to employ this technique; a sentence or two explaining their rationale would be of interest to other structural labs interested in studying similarly sized macromolecular complexes.

2) A final summarizing cartoon outlining the relationship of Lis1 to the mechanochemical cycle of the dynein motor, and how it translates to MT affinity would strengthen the impact of this manuscript.

3) No mention is made of deposition of the 5 EM densities to the EMDB. It is important that these are made available to the community.

4) The authors state in their Abstract that how Lis1 regulates dynein is unknown, but in light of their previous work as well as that of the Vallee lab, this is a clear over-statement. This continues on through the Introduction, where they underplay their previous work determining the binding interface of Lis1 on dynein. This should be revised.

5) The authors set up an either/or model that seems overly simplistic, and the terms ring-centric and linker-centric are clunky and not very clear.

6) The authors report that their dynein alone map “accommodates the crystal structure of the dynein motor domain well”. While there is an overall match, there are also clearly discrepancies between the structures. How was the fit assessed quantitatively? Can anything be learned from the discrepancies?

7) The authors report that the 3D structure allows them “to hone in on the region of dynein interacting with Lis1”, but other than the displacement of the linker, they do not go further with this approach. Are there any other changes? It would be good to discuss this, and the Lis1-dynein interface in more detail.

8) It is unclear what new is learned from the section on “one dynein ring binds one Lis1 beta-propeller” Isn't this what they studied in their previous paper?

9) Did they intend to reference Carragher et al?

10) Vanadate is not so much a phosphate analog as a transition state analog. Similarly, calling the ADP-V complex a “dead-end” is awkward and imprecise terminology.

11) The authors suggest that the differences in AAA4 and AAA5 linker positions previously noted might correspond to either distinct mechanochemical states or different dynein species. But isn't it also possible that they may result from different constructs used in the different studies, some of which were truncated or chimeric?

12) Figure 2 would be stronger if the authors included parallel studies with the W419A mutant and provided more complete analysis of the motility data. Based on what is shown here, the R378A mutant effectively blocks formation of the dynein-Lis1 complex as judged by gel filtration, but only partially blocks the effects of Lis1 on dynein motility. Shoring up this point with an additional single point mutation would strengthen the point.

13) The authors report that two structures are seen in the presence of ADP and one in the presence of no nucleotide, so they conclude that the AAA4 linker position is the ADP state. This may be true, but kinetic analysis has indicated that there is isomerization between 2 ADP-bound states, so it is also possible that the two observed structures correspond to different ADP-bound states. Both should be shown in Figure 3. Didn't the authors perform FRET analysis of the dynein-Lis1 complex alone? How come this is not included in Figure 3?

14) The authors report the ADP-V-dynein-Lis1 complex. Wouldn't it be useful to directly compare this to the ADP-V-dynein complex?

15) The experiments described in the text and in Figure 4 yield results that are predictable. Three papers, including work from this same group, have already shown that Lis1 does not significantly affect dynein's ATPase rate, so it is unclear what new is learned from this work. It could all be removed from the text, or at least moved to supplemental information. If it does remain, experiments need to be performed in triplicate, and error bars should be reported for all points.

16) In Figure 5, for the short linker construct, why are there fewer events, including fewer rebinding events? Does this construct have a weaker affinity for microtubules? If it does, how might this affect the interpretation of the results?

17) Use of dimeric Lis1. It is not clear how an avidity effect would explain how dimeric Lis1 would bind more tightly to monomeric dynein than monomeric Lis1. The avidity effect should require two binding sites for Lis1. In Huang et al there does not seem to be a direct demonstration that dimeric Lis1 binds monomeric dynein better than monomeric Lis1. Please clarify.

18) If the Lis1 purely displaces the linker by a steric mechanism, would the authors not expect to see the linker closely connected to Lis1? Instead Figure 1 suggests that the linker is making a tight connection to the AAA+ ring. Should this connection be discussed? Might it suggest that Lis1 has some allosteric effect on the AAA+ ring which, together with sterically blocking its normal position, allows the linker to dock onto the ring? Alternatively is the resolution too low to say this?

19) The authors try to draw the parallel between mutations in AAA5 and the effect of Lis1. The reviewer's reading of the data is that these two effects are different as the first abolishes ATPase activity, while the second has no effect on it.

20) One section states that the lack of conservation in the very N-terminus of the linker suggests it does not interact directly with Lis1 (which seems reasonable) and therefore that the Lis1 is acting as a steric block. The contact between the linker and the AAA+ ring in the Lis1:APO-Dynein structure appears to involve a more conserved part of the ring. This comes back to the question about whether Lis1 is acting on the AAA+ ring to allow the linker to bind to a place it normally would not.

21) Figure 2. The blue color is difficult to distinguish from the black in the gel filtration trace.

22) Figure 3/g. This EM map looks as though the contacts between the AAA+ ring and the Lis1 are different from in the Apo form. This may just be the different resolution of the structure, but it might be useful to discuss this somewhere.

23) Figure 4. It would be helpful if the type of mutation made was specified on this figure. b & c are E-Q mutations (residue number?). d) AAA5 mutant is not in the ATP binding site, but on the linker binding site. Specifying this on the figure (& res numbers?) would be helpful.

24) Figure 5 and Figure 5—figure supplement 1. These figures show release from microtubules is blocked by Lis1, but not if the linker is shortened. The shortening does not affect the motors activity or ability to bind Lis1. Did the authors test if the velocity of the shorter linker was affected by Lis1?

25) Figure 5 d ii: The cartoon uses the stop sign, which suggests it has a mutation in AAA5 (as in Figure 4). Is this true? If not then another symbol might be more appropriate.

26) Table 3:

The units of Km, kbasal and kcat should be given.

One of the reviewers has the following thought which they would like to share with you:

Another mechanism that could be mentioned in the Discussion is that Lis1 has an allosteric effect on the AAA+ ring that allows the linker to move and the AAA1 site to exchange nucleotide without triggering the conformational change in the stalk that releases dynein from microtubules. This model is also not completely satisfactory as it is harder to explain the requirement for the N-terminus of the linker.

Perhaps one (explanation would be that the transition from high to low affinity is most likely to occur before the linker moves from a post to pre-powerstroke position. That is, ATP binds to a high affinity, post-powerstroke dynein. A rearrangement occurs that leads to a low affinity, post-powerstroke dynein (that releases from MTs). This is followed by a transition to the ADP.Vi like state with a low affinity, pre-powerstroke conformation. The ATP bound, post-powerstroke, low affinity conformation could be altered so that it no longer releases from microtubules by a combination of Lis1 AND linker binding to the AAA+ ring.

---

## [Author Response]

*1) There is some concern that much of this work follows directly from the outstanding*
[23]
*paper published by this group. In this paper, the authors further refine the model developed there, and while the three-dimensional structure adds knowledge, it is limited in detail because the docking of Lis1 is not fully resolved in terms of rotational alignment. The couple of new mutants examined here could also have been predicted based on the previous structure*.

*The most novel idea here is that the binding of Lis1 provides a steric block to the linker swing. This is tested with a single construct with a truncated linker, and the results support their model. But there is concern that this is a single approach; the paper would have more impact if the authors could demonstrate this key point in a complementary way. The reviewers recognize that this might be a tall order, but it would be good if the authors could respond to this issue*.

To further test the steric block model of Lis1 regulation of dynein we chose to explore if the specific sequence of the N-terminal portion of the linker (the part that is removed in the short linker dynein) is important for Lis1 regulation of dynein (as opposed to its simply being a linker-like protein mass that can be blocked). We chose two different sequences to add back to the short linker dynein to restore the wild type linker length: (1) The sequence from *S. pombe*, which appears to lack a Lis1 homolog; and (2) The sequence from an axonemal dynein from *C. reinhardtii* (dynein-c), which does not bind axonemal Lis1. Our goal was to then use single-molecule microtubule binding/release assays on these chimeric dyneins, alone and in the presence of Lis1, to test if Lis1 regulation was restored. Unfortunately, the chimeric axonemal dynein failed to express. The chimeric *S. pombe* dynein did not express well, either, but we were able to purify amounts that were sufficient for single-molecule studies. However, this chimeric dynein showed a disrupted mechanochemical cycle: it was unable to release from microtubules in the presence of ATP even in the absence of Lis1. We agree that additional evidence for the steric block hypothesis would have strengthened our model; unfortunately, given the results described above, this does seem to be too tall an order to be accomplished in a reasonable time frame.

*2) The authors interpret their data with a model in which the linker needs to get to AAA5 in order for the cycle of ATP hydrolysis to continue and so blocking access to AAA5 prevents ATP binding and therefore release from microtubules (which is thought to occur on ATP hydrolysis). The problem with this model, as the authors discuss, is that it would predict that Lis1 would inhibit the dynein ATPase. The data, however, are clear that this is not the case. It is suggested that the authors reduce the emphasis on the model that Lis1 blocks the linker from getting to AAA5. Instead the data could be presented and then this discussed as an option*.

We agree with this suggestion and have made the appropriate changes in the last Results section and the Discussion.

More specific points to consider:

*1) The authors employed cryo-negative staining for their EM analysis. This involved and tedious technique is not commonly used by EM labs, as it is difficult to master. Surely the authors had a reason to employ this technique; a sentence or two explaining their rationale would be of interest to other structural labs interested in studying similarly sized macromolecular complexes*.

Thank you for this suggestion; we have added text to both the Results and Methods sections to explain our choice.

*2) A final summarizing cartoon outlining the relationship of Lis1 to the mechanochemical cycle of the dynein motor, and how it translates to MT affinity would strengthen the impact of this manuscript*.

We have added a model figure (Figure 6).

*3) No mention is made of deposition of the 5 EM densities to the EMDB. It is important that these are made available to the community*.

We apologize for accidentally not making our intent regarding EMDB submission clear. Final maps as well as raw half maps for each reconstruction have been deposited with the EMDB. This is now also stated in the manuscript.

*4) The authors state in their Abstract that how Lis1 regulates dynein is unknown, but in light of their previous work as well as that of the Vallee lab, this is a clear over-statement. This continues on through the Introduction, where they underplay their previous work determining the binding interface of Lis1 on dynein. This should be revised*.

We’ve discussed this point extensively among us and feel we disagree with this assessment. What we state in our Abstract is that “The dynein regulator Lis1 is known to keep dynein bound to microtubules; however, how this is accomplished mechanistically remains unknown”. We feel this statement is correct; the key distinction here is between “what” Lis1 does (which we established in our 2012 paper), and “how” it does it (which we did not). Our Introduction summarizes how much is known, from both our and the Vallee lab’s work, about what Lis1 does to dynein. However, we feel there is a clear difference between what are ultimately descriptions of biochemical effects from a more detailed understanding of the mechanism behind them. While we have not yet answered all the questions in this regard, we think the work presented here has gone a long ways towards explaining the “how”.

Similarly, our previous work did not address the binding site on Lis1 for the dynein-Lis1 interaction but rather only residues in dynein. Some other important aspects of the dynein-Lis1 interface that were not addressed by our previous work include the face of the ring to which Lis1 binds (which would determine whether the linker might be directly affected), and which side of the Lis1 propeller interacted with dynein (even if sequence conservation had already made a strong argument for what we saw in our structure).

*5) The authors set up an either/or model that seems overly simplistic, and the terms ring-centric and linker-centric are clunky and not very clear*.

Thank you for the comment. We now use “steric” and “allosteric” models, and do not present them as being mutually exclusive.

6) The authors report that their dynein alone map “accommodates the crystal structure of the dynein motor domain well”. While there is an overall match, there are also clearly discrepancies between the structures. How was the fit assessed quantitatively? Can anything be learned from the discrepancies?

We have carried out a Fourier Shell Correlation (FSC) between the dynein alone map and the crystal structure of the dynein motor domain (PDB ID: 4AKG). We see an FSC of 0.143 at a resolution of 18.8 Å. We deposited an XML file of the FSC plot in the EMDB as a supplementary file to the dynein alone submission (EMDB-6013). This is briefly mentioned in Results and the deposition is noted in the Accession Numbers section in Materials and methods.

With regards to discrepancies between the EM map and the crystal structure in some regions, we felt a rigid body fit of the entire motor domain into the density was most appropriate at the resolution of our map. Although we are very interested in whether these differences exist and what they might be, we feel we do not yet have the resolution that would be required to fit individual AAA+ motifs in the density (and comment on that fit).

*7) The authors report that the 3D structure allows them “to hone in on the region of dynein interacting with Lis1”, but other than the displacement of the linker, they do not go further with this approach. Are there any other changes? It would be good to discuss this, and the Lis1-dynein interface in more detail*.

The displacement of the linker is the most striking difference in the dynein-Lis1 map compared to that of dynein alone. We are also very curious to see whether there are further changes in dynein’s ring upon Lis1 binding and will pursue this in future work. However, we feel that at the current ∼21 Å resolution we cannot trust, and comment on, any changes less apparent than the displaced linker domain.

With respect to the Lis1-dynein interface, we were able to directly visualize that dynein contacts Lis1 via the AAA3-AAA4 connector helix. Unfortunately, as we state in the paper, we did not have the resolution to determine Lis1’s rotational fit in the density. We therefore feel that we cannot currently go further than the view of the interface we present in Figure 2. We worry that any additional details might lead readers into assuming the interface is fully established.

8) It is unclear what new is learned from the section on “one dynein ring binds one Lis1 beta-propeller” Isn't this what they studied in their previous paper?

We thank the reviewers for bringing this up, as we did not clearly highlight what was new in our current findings.

In our previous study, we used 2D classification of dimeric Lis1 in complex with a dimeric dynein. We observed extra density in a difference map that we could assign to Lis1 (thus establishing where on dynein Lis1 bound) but we couldn’t determine whether the density accommodated one or two propeller domains, or indeed any of the N-terminal domain of Lis1. It was not until we had the 3D map reported here that we could resolve the channel at the center of the propeller and thus were able to unambiguously show that a single propeller (and that domain alone) contacts the dynein motor domain.

We have modified this section of the Results to make the new findings stand out more clearly.

9) Did they intend to reference Carragher et al?

Thank you for spotting this error, we have replaced it with the correct reference.

*10) Vanadate is not so much a phosphate analog as a transition state analog. Similarly, calling the ADP-V complex a “dead-end” is awkward and imprecise terminology*.

We now refer to ADP.Vi as a transition state analog and have removed the term “dead-end”. Thank you for these suggestions.

11) The authors suggest that the differences in AAA4 and AAA5 linker positions previously noted might correspond to either distinct mechanochemical states or different dynein species. But isn't it also possible that they may result from different constructs used in the different studies, some of which were truncated or chimeric?

Thank you for pointing this out; we have added this possibility to the text.

*12)*
Figure 2
*would be stronger if the authors included parallel studies with the W419A mutant and provided more complete analysis of the motility data. Based on what is shown here, the R378A mutant effectively blocks formation of the dynein-Lis1 complex as judged by gel filtration, but only partially blocks the effects of Lis1 on dynein motility. Shoring up this point with an additional single point mutation would strengthen the point*.

To give a more detailed account of the effects of the Lis1 mutants on dynein velocity we have added a histogram of velocity distributions for the motility experiments (new figure; Figure 2—figure supplement 2). We have not added additional analysis of run length data because we have previously found that velocity analysis is the most sensitive measure of Lis1’s effect on dynein. Run length analysis provides little information; since Lis1 both slows dynein down and prolongs its microtubule encounters, which have opposing effects on run length, its addition results in little change in the average travel distance [23].

With regards to the interpretation of the R378A mutant and analysis of mutant W419A, gel filtration is not a very sensitive measure of the dynein-Lis1 interaction; we would not expect a weak interaction to be detected in this assay. This is why we stated in the paper that “binding is severely weakened” for the R378A mutant rather than that this interaction does not exist (particularly given that the motility data still show that it does, albeit in a limited way). While doing the experiments with the W419A mutant suggested was feasible, these experiments would have been fairly time-consuming as the yeast strains needed for this analysis were not already available. Since we were not confident that the time investment would yield new information (given that this mutation is already encompassed in the Lis1^5A^ quintuple mutant), we chose to invest our time in experiments that would test the major mechanistic hypothesis of the paper, as suggested in general point #1 (even if those experiments were unsuccessful in the end). We have reworded the manuscript to address the apparent discrepancy between the gel filtration and single-molecule assays for R378A, making it clear that the single-molecule assays are more sensitive in terms of detecting an interaction and that not seeing co-migration by gel filtration is not an indication of a complete abolishment of that interaction.

*13) The authors report that two structures are seen in the presence of ADP and one in the presence of no nucleotide, so they conclude that the AAA4 linker position is the ADP state. This may be true, but kinetic analysis has indicated that there is isomerization between 2 ADP-bound states, so it is also possible that the two observed structures correspond to different ADP-bound states. Both should be shown in*
Figure 3*. Didn't the authors perform FRET analysis of the dynein-Lis1 complex alone? How come this is not included in*
Figure 3*?*

Thank you for pointing this out, we have added the AAA4 linker position structure and FRET analysis of dynein-Lis1 (no nucleotide conditions) to Figure 3.

*14) The authors report the ADP-V-dynein-Lis1 complex*. *Wouldn't it be useful to directly compare this to the ADP-V-dynein complex?*

We are indeed very curious to see the structure of ADP.Vi-dynein (in the absence of Lis1) and compare it to the one we reported for the complex (ADP.Vi-dynein-Lis1) but feel this is beyond the scope of the current paper. The goal of the ADP.Vi-dynein-Lis1 structure was simply to establish (visually, in this case) whether the linker domain could reach AAA2 in the presence of Lis1.

*15) The experiments described in the text and in*
Figure 4
*yield results that are predictable. Three papers, including work from this same group, have already shown that Lis1 does not significantly affect dynein's ATPase rate, so it is unclear what new is learned from this work. It could all be removed from the text, or at least moved to supplemental information. If it does remain, experiments need to be performed in triplicate, and error bars should be reported for all points*.

While it has indeed been shown before that Lis1 does not significantly affect dynein’s ATPase rate, no one had addressed the question of which AAA+ domain was responsible for the continued ATP turnover. In the work we report here we measured, for the first time, what happens to ATP hydrolysis in the presence of Lis1 when AAA1 or AAA4 are rendered hydrolysis deficient or AAA5 is linker-docking deficient. The experiments we present are therefore the first to probe which of dynein’s AAA+ domains are responsible for the observed nucleotide hydrolysis in the presence of Lis1 and, for that reason, we feel they belong in the main text.

Regarding the error bars, as the number of experiments was ≤ 3 (triplicate for experiments showing an activated ATPase rate, and duplicate for those that are ATPase deficient), we plotted each of these points separately. Error bars are not robust and can be misleading for such sample sizes [Krzywinski & Altman (2013) Nature Methods 10, 921–922]. Plotting individual data points is recommended practice in this situation, as it explicitly shows the spread in the data.

*16) In*
Figure 5*, for the short linker construct*, *why are there fewer events, including fewer rebinding events? Does this construct have a weaker affinity for microtubules? If it does, how might this affect the interpretation of the results?*

The full-length and short linker constructs appear to have a similar affinity for microtubules based on the similar concentration of microtubules required for half-maximal activation of their ATPase rates [K_m_(MT)] (Table 3). The density of the full-length and short linker constructs on the microtubule in the release assays is also similar (0.289 ± 0.080 and 0.226 ± 0.081 dynein molecules per micrometer of microtubule, respectively). Small differences in density likely arise from different dilutions used. Regarding rebinding events, apart from cases where Lis1 triggers a clear persistent microtubule attachment state in dynein (i.e. full-length linker dynein with Lis1), comparative measurements are not very meaningful as rebinding events are in part dependent on the buffer flow rate through each flow chamber, which guides the rate of clearance of dynein molecules released from microtubules. This flow rate is manually driven via a syringe; so, it too can vary slightly between assays. Therefore, we do not base any of our conclusions on rebinding frequency. In summary, the data suggest that the short linker dynein has a similar microtubule affinity as the full-length linker dynein, and so our interpretations are not affected.

*17) Use of dimeric Lis1. It is not clear how an avidity effect would explain how dimeric Lis1 would bind more tightly to monomeric dynein than monomeric Lis1. The avidity effect should require two binding sites for Lis1. In Huang et al there does not seem to be a direct demonstration that dimeric Lis1 binds monomeric dynein better than monomeric Lis1. Please clarify*.

We apologize for the lack of clarity on this point. Previously we showed that a much greater concentration of monomeric Lis1 is needed to achieve the same effect on dynein motility observed with dimeric Lis1 [23]. This is the reason why we chose to use dimeric Lis1 in our EM work: it allowed us to use a much lower concentration of it, keeping the background in the images low. We have changed the wording for this section in the Results to discuss this in terms of local Lis1 β-propeller concentration rather than avidity.

*18) If the Lis1 purely displaces the linker by a steric mechanism, would the authors not expect to see the linker closely connected to Lis1? Instead*
Figure 1
*suggests that the linker is making a tight connection to the AAA+ ring. Should this connection be discussed? Might it suggest that Lis1 has some allosteric effect on the AAA+ ring which, together with sterically blocking its normal position, allows the linker to dock onto the ring? Alternatively is the resolution too low to say this?*

It is indeed difficult at the resolution of the dynein-Lis1 map to discriminate between physical proximity to the AAA+ ring and a specific interaction. However, the fact that the linker is flexible in its position relative to the ring (Figure 1—figure supplement 1) we think argues against a tight interaction; in that case, we would have expected to be able to resolve the linker at AAA4 without having to use 3D sorting.

Despite this circumstantial evidence, we wanted to test a possible linker-AAA4 interaction directly. We mutated 5 residues on AAA4 that appear, based on our map, to be good candidates for a contact with the linker (new figure added; Figure 1—figure supplement 2). While this mutant (where all 5 residues were changed to alanine) showed a statistically significant difference in the extent to which Lis1 reduced its velocity relative to wild type, the difference was small and the AAA4 mutant was still regulated by Lis1. Taken together, the EM analysis and mutagenesis data argue strongly against a specific interaction between the linker and AAA4.

*19) The authors try to draw the parallel between mutations in AAA5 and the effect of Lis1. The reviewer's reading of the data is that these two effects are different as the first abolishes ATPase activity, while the second has no effect on it*.

While it is true that there is a difference in the ATPase activity between the two effects, there are also two similarities; a decrease in dynein velocity and a persistent attachment to the microtubule. We therefore feel that the comparison is useful in terms of trying to understand how Lis1 brings about a slow-moving, microtubule-attached dynein. However, we do agree that Lis1’s effect on the ATPase rate is still very much an open question, perhaps involving a secondary mechanism of allostery through the AAA+ ring. We have added new text to the Discussion to address this unresolved question.

*20) One section states that the lack of conservation in the very N-terminus of the linker suggests it does not interact directly with Lis1 (which seems reasonable) and therefore that the Lis1 is acting as a steric block. The contact between the linker and the AAA+ ring in the Lis1:APO-Dynein structure appears to involve a more conserved part of the ring. This comes back to the question about whether Lis1 is acting on the AAA+ ring to allow the linker to bind to a place it normally would not*.

We have addressed this in point 18 above.

*21)*
Figure 2*. The blue color is difficult to distinguish from the black in the gel filtration trace*.

We thank the reviewers for pointing this out. We have made the purple and green traces lighter to set them apart from the black more clearly.

*22)*
Figure 3*/g. This EM map looks as though the contacts between the AAA+ ring and the Lis1 are different from in the Apo form. This may just be the different resolution of the structure, but it might be useful to discuss this somewhere*.

It is likely that the AAA+ ring undergoes structural rearrangements in the ATP.Vi state that could change the position of Lis1’s binding site. However, as we pointed out, the resolutions of the two structures (apo vs. ATP.Vi) are different; since the ATP.Vi map is the lowest resolution structure we present we do not feel confident about commenting on possible differences in the AAA+ ring and/or contacts with Lis1.

*23)*
Figure 4*. It would be helpful if the type of mutation made was specified on this figure. b & c are E-Q mutations (residue number?). d) AAA5 mutant is not in the ATP binding site, but on the linker binding site. Specifying this on the figure (& res numbers?) would be helpful*.

We thank the reviewers for the suggestion. All mutations and residue numbers were originally given in the figure legend because we felt that adding them directly to the figure itself would clutter it, particularly since the AAA5 linker-docking mutant is a triple mutation. In the current version we have modified the headings for each plot to make the mutation type clear without making reading the figure legend necessary.

*24)*
Figure 5
*and*
Figure 5—figure supplement 1. *These figures show release from microtubules is blocked by Lis1, but not if the linker is shortened. The shortening does not affect the motors activity or ability to bind Lis1. Did the authors test if the velocity of the shorter linker was affected by Lis1?*

It is not possible to test the effect of Lis1 on velocity in the context of a monomer gliding assay because it is known that the short linker dynein monomer moves slowly in gliding assays ([47], Figure 3), but moves with wild type speeds as a dimer in single-molecule assays ([47], Figure 3). The slow gliding by the short linker monomer is likely due to motor-glass interactions. Examining how Lis1 affects the velocity of dimeric short linker dynein is also problematic; once the short linker dynein is dimerized with GST its linker is no longer “short” (as it is extended by GST) and its ability to bypass Lis1 is now blocked by the dimerization.

*25)*
Figure 5
*d ii: The cartoon uses the stop sign, which suggests it has a mutation in AAA5 (as in*
Figure 4*). Is this true? If not then another symbol might be more appropriate*.

Thank you for pointing out the potential confusion. We have now moved all references to the linker-AAA5 interaction to the Discussion and new model figure (Figure 6). Therefore the stop sign in this figure has been removed.

*26)*
Table 3*:*

*The units of Km, kbasal and kcat should be given*.

The units have been added.

One of the reviewers has the following thought which they would like to share with you:

*Another mechanism that could be mentioned in the Discussion is that Lis1 has an allosteric effect on the AAA+ ring that allows the linker to move and the AAA1 site to exchange nucleotide without triggering the conformational change in the stalk that releases dynein from microtubules. This model is also not completely satisfactory as it is harder to explain the requirement for the N-terminus of the linker*.

*Perhaps one (explanation would be that the transition from high to low affinity is most likely to occur before the linker moves from a post to pre-powerstroke position. That is, ATP binds to a high affinity, post-powerstroke dynein. A rearrangement occurs that leads to a low affinity, post-powerstroke dynein (that releases from MTs). This is followed by a transition to the ADP.Vi like state with a low affinity, pre-powerstroke conformation. The ATP bound, post-powerstroke, low affinity conformation could be altered so that it no longer releases from microtubules by a combination of Lis1 AND linker binding to the AAA+ ring*.

We agree that dynein is normally likely to detach from the microtubule before the linker moves to AAA2 (there is kinetic evidence that this is the case in *Dictyostelium* dynein [24]), and that the transition from the high-to-low microtubule affinity (before the linker has moved) is the most-likely transition to be regulated by Lis1. However, our short linker experiments do suggest that steric regulation via the linker is critical for this regulation (i.e. it cannot be purely allosteric).